# ATPase activity of DFCP1 controls selective autophagy

Viola Nähse ®[1,2,3] ✉, Camilla Raiborg ®[1,2], Kia Wee Tan[1,2,5], Sissel Mørk[1,2], Maria Lyngaas Torgersen[1,2], Eva Maria Wenzel[1,2], Mireia Nager[4], Veijo T. Salo ®[3,6], Terje Johansen ®, Elina Ikonen ®[3], Kay Oliver Schink ®[1,2,7] ✉ & Harald Stenmark ®[1,2] ✉

Cellular homeostasis is governed by removal of damaged organelles and protein aggregates by selective autophagy mediated by cargo adaptors such as p62/SQSTM1. Autophagosomes can assemble in specialized cup-shaped regions of the endoplasmic reticulum (ER) known as omegasomes, which are characterized by the presence of the ER protein DFCP1/ZFYVE1. The function of DFCP1 is unknown, as are the mechanisms of omegasome formation and constriction. Here, we demonstrate that DFCP1 is an ATPase that is activated by membrane binding and dimerizes in an ATP-dependent fashion. Whereas depletion of DFCP1 has a minor effect on bulk autophagic flux, DFCP1 is required to maintain the autophagic flux of p62 under both fed and starved conditions, and this is dependent on its ability to bind and hydrolyse ATP. While DFCP1 mutants defective in ATP binding or hydrolysis localize to forming omegasomes, these omegasomes fail to constrict properly in a size-dependent manner. Consequently, the release of nascent autophagosomes from large omegasomes is markedly delayed. While knockout of DFCP1 does not affect bulk autophagy, it inhibits selective autophagy, including aggrephagy, mitophagy and micronucleophagy. We conclude that DFCP1 mediates ATPase-driven constriction of large omegasomes to release autophagosomes for selective autophagy.

Autophagy is a cellular degradation mechanism that is critical to maintain cellular homeostasis – either by recycling of bulk cytoplasmic components to produce energy in times of starvation or by degradation of potentially harmful cytoplasmic objects. During autophagy, cargo is engulfed by a double membrane, the phagophore. After sealing of the phagophore, the autophagosome – containing the engulfed cargo – fuses with a lysosome for degradation of its content[1].

The ER is considered the major site of autophagosome formation[2,3]. A major regulator of autophagy is the class III phosphatidylinositol 3-kinase complex (PI3K-III), which is activated by pro-autophagic cues to generate phosphatidylinositol 3-phosphate (PtdIns3P) at specialized regions of the ER. These PtdIns3P-rich ER-subdomains serve as platforms for autophagosome assembly. They form characteristic cup-like structures, which are called

[1]Centre for Cancer Cell Reprogramming, Faculty of Medicine, University of Oslo, Montebello N-0379 Oslo, Norway. [2]Department of Molecular Cell Biology, Institute for Cancer Research, Oslo University Hospital, Montebello 0379 Oslo, Norway. [3]Department of Anatomy and Stem Cells and Metabolism Research Program, Faculty of Medicine, University of Helsinki, Helsinki, Finland. [4]Autophagy Research Group, Department of Medical Biology, University of Tromsø, The Arctic University of Norway, Tromsø, Norway. [5]Present address: Department of Medical Cell Biology, Uppsala University, Uppsala, Sweden. [6]Present address: Structural and Computational Biology Unit, European Molecular Biology Laboratory, Heidelberg, Germany. [7]Present address: Department of Molecular Medicine, Institute of Basic Medical Sciences, University of Oslo, PO Box 1112 Blindern 0317 Oslo, Norway. ✉e-mail: Viola.Naehse@rr-research.no; k.o.schink@medisin.uio.no; h.a.stenmark@medisin.uio.no

'omegasomes'[4,5]. Within the omegasome cup, a double membrane, the phagophore, is formed, which extends and engulfs the cargo. During this process, membrane contact sites with the ER likely provide the membrane for the extension of the phagophore[5,6]. Once the phagophore has engulfed the cargo, it closes into an autophagosome and dissociates from the ER[5,7].

In line with the requirement for PtdIns3P, several PtdIns3P-binding proteins are important for autophagosome formation and closure. During early phases of the phagophore formation, two groups of PtdIns3P-binding proteins are recruited. Proteins of the WIPI (WD-repeat protein interacting with phosphoinositides) family[8] are necessary for lipid conjugation and activation of the phagophore-forming protein LC3B[9,10]. The other PtdIns3P-binding protein, DFCP1 (Double FYVE containing protein 1)[4] is recruited to omegasomes. While DFCP1 has been widely used as an early autophagy reporter, its cellular functions are largely unknown, and its role during autophagosome formation is not understood.

Here, we show that DFCP1 is a large, Dynamin-related ATPase that dimerizes in an ATP-dependent fashion at specific ER sites. We further demonstrate that the ability of DFCP1 to bind and hydrolyze ATP is required for proper omegasome constriction and selective autophagy.

## Results

### DFCP1 binds and hydrolyzes ATP

The main structural features of DFCP1 are two C-terminal FYVE domains, which, in conjunction with an ER-targeting region, bind to PtdIns3P on ER domains and by this mark the sites of autophagosome formation[4] (Fig. 1a). In contrast, the N-terminus of DFCP1 does not carry any characterized domains, but we found it interesting that it contains a P-Loop motif[4]. A search in the Interpro databases revealed that the DFCP1 domain structure, including the P-loop, the ER-binding domain and the two FYVE domains, is an ancient architecture which evolved more than 500 million years ago, with homologs in most metazoan phyla (Supplementary Fig. S1a, b). P-loops interact with phosphorylated nucleotides, but no function of the N-terminal domain has so far been described for DFCP1.

Structure prediction using AlphaFold[11], as well as structural homology modeling using the Phyre2 server[12] revealed that the DFCP1 P-Loop domain shows structural homology to large nucleotide-binding proteins, especially to the GTPases Atlastin and GBP1 (Fig. 1b, Supplementary Fig. S1c), suggesting that it could be a nucleotide-binding protein. To test this hypothesis, we tested if DFCP1 is able to bind nucleotides. We purified the N-terminus of DFCP1, including the predicted nucleotide-binding domain (Supplementary Fig. S2a). To measure nucleotide binding, we used N-methylanthraniloyl (mant)-modified nucleotides, mantATP and mantGTP, which are weakly fluorescent in solution but strongly increase fluorescence upon binding to a protein. We observed no binding to mantGTP within the time course of our measurements, whereas Cdc42 – a GTP-binding protein – showed robust binding to mantGTP (Fig. 1d). In contrast, DFCP1 efficiently and specifically bound to ATP and to ADP (Fig. 1c), suggesting that it could act as a molecular switch, depending on the loaded nucleotide. Binding of ATP or ADP to DFCP1 was rapid in comparison with Cdc42, with the reaction reaching saturation within minutes (Fig. 1c, d). Most Dynamin-related NTPases have low or moderate nucleotide-binding affinities in the μM range (Dynamin-1 μM[13], Atlastin 2.15 μM[14], mEHD2 13 μM[15]). To determine the ATP-binding affinity of DFCP1, we performed equilibrium binding assays using constant levels of mantATP/mantADP and varying concentrations of DFCP1. These assays showed also that DFCP1 has a relatively low nucleotide affinity, with an apparent equilibrium dissociation constant ($K_D$) of 10.7 μM for ATP binding and a slightly higher affinity for ADP (5.4 μM) (Fig. 1e). This relatively low affinity is in line with the observed fast nucleotide exchange rates.

To address whether DFCP1 has ATPase activity, we measured DFCP1-dependent phosphate release from ATP. Indeed, incubation of the purified DFCP1 ATPase domain with ATP resulted in the release of free phosphate (Fig. 1f). Thus, DFCP1 is a functional ATPase.

To understand the functional importance of the ATP binding and hydrolysis by DFCP1, we aimed to generate mutations that affect these biochemical properties. We used a Phyre2-generated homology model based on Atlastin, which showed the highest degree of sequence similarity (Fig. 1a, b). Using this model, we performed structural alignments with other nucleotide-binding proteins (Atlastin, GBP1, N-RAS and K-RAS) to identify key residues necessary for nucleotide binding and hydrolysis.

NTPases have a characteristic GKS motif, which is critical for nucleotide binding by coordinating a magnesium ion[16]. We identified this motif at residues 192–194 of DFCP1. Based on homologies with the small GTPases K-RAS and N-RAS, we further identified the residues T189 and G190 as potentially critical residues. T189 is at the same position as the G12 residue of K-RAS, whereas G190 corresponds to G13 in K-RAS. Both amino acids are critical for K-RAS GTPase activity (Fig. 1b). To test if these residues can contact a bound nucleotide, we used an Alphafold-generated model of DFCP1 and used structural alignments to model the localization of a nucleotide (GTP) based on the crystal structure of GBP1 (Fig. 1g) using the "Matchmaker" function of UCSF ChimeraX. Inspection of the resulting model showed that the identified residues are well-placed to contact a bound nucleotide (Fig. 1g, inset).

Based on these predictions, we generated and purified variants of the DFCP1 ATPase domain with mutations in the nucleotide-binding residues (Supplementary Fig. S2a). Similar to other nucleotide-binding proteins, mutation of the GKS motif (K193A, S194N) resulted in a complete loss of ATP binding (Fig. 1h). The same was the case with mutation G190V, with G190 being analogous to G13 of K-RAS (Fig. 1h).

Next, we aimed to identify an ATPase-defective DFCP1 mutant. We found that the mutation T189V, which corresponds to the hydrolysis-defective K-Ras G12V allele, showed slightly reduced hydrolysis activity, but showed also reduced nucleotide binding (Fig. 1h, i). However, another mutation of T189, T189A, showed robust nucleotide binding but strongly reduced ATP-hydrolysis activity (~1/4 of the wild-type protein) (Fig. 1j, k), thus likely constituting a primarily ATP-bound DFCP1 allele.

### DFCP1 ATPase activity is stimulated by membrane binding

Many Dynamin-related NTPases not only contain a nucleotide-binding domain, but also membrane binding domains such as PH domains. Moreover, Dynamin GTPases and EHD-family ATPases show increased nucleotide hydrolysis rates in the presence of membranes[15,17,18].

As DFCP1 has two PtdIns3P-binding FYVE domains and localizes to ER-associated omegasomes, we speculated also that nucleotide hydrolysis of DFCP1 might be influenced by the presence of membranes. To test this hypothesis, we purified full-length recombinant DFCP1 from insect cells and tested if the presence of membranes resulted in increased ATP hydrolysis. We used liposomes (100 nm diameter) mimicking the ER and doped with PtdIns3P or PI. Addition of liposomes to purified full-length DFCP1 resulted in a modest, but significant increase in ATP hydrolysis (~1.4 fold) over an assay time of 30 min, indicating that DFCP1 ATPase activity is indeed stimulated by membranes (Fig. 2a).

### DFCP1 dimerizes upon ATP binding

Many NTPases, such as Dynamin and Atlastin, dimerize in a NTP-dependent fashion[19]. The dimerization interface is a highly conserved region in the NTPase domain, the "G-interface", which includes the nucleotide-binding site[19]. To experimentally test if DFCP1 shows nucleotide-dependent dimerization, we loaded the purified DFCP1 N-terminus with ADP or the non-hydrolysable ATP analog, ATPγS, and

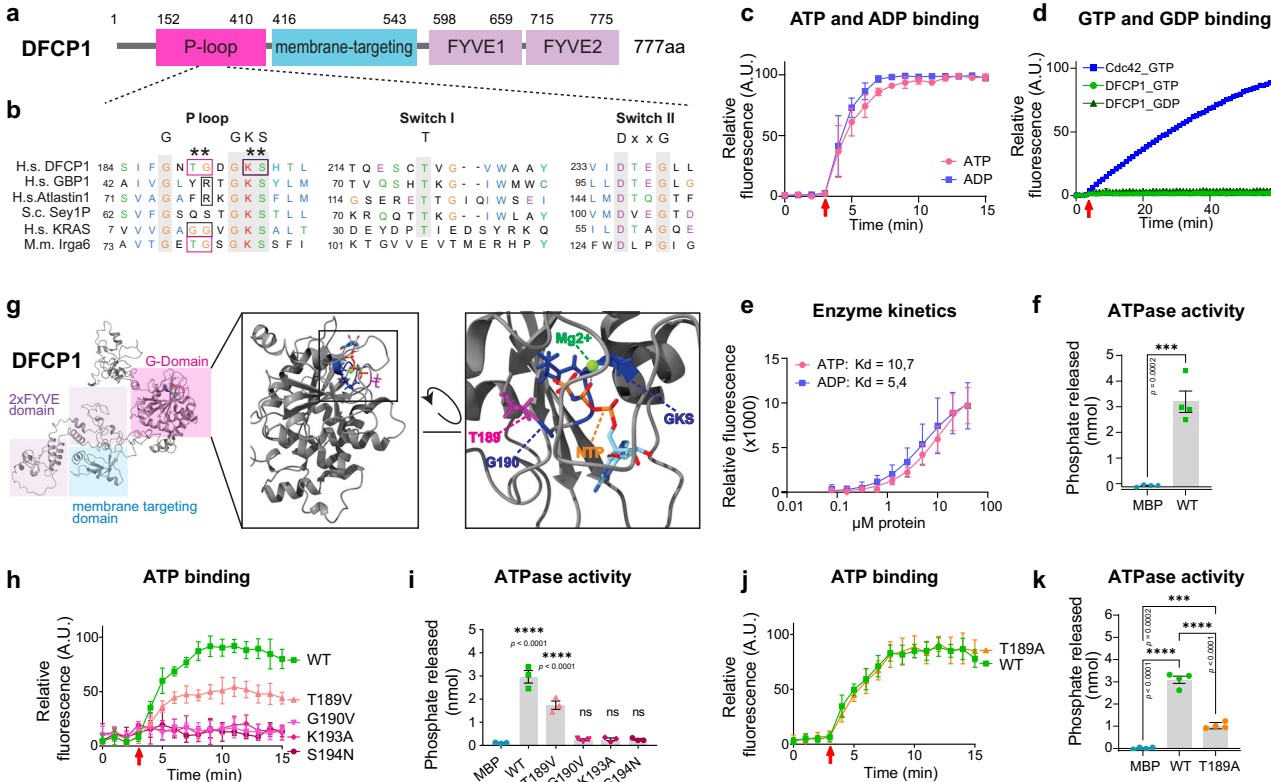

**Fig. 1 | DFCP1 has structural similarities to ATPases and GTPases and binds and hydrolyzes ATP. a** Domain structure of DFCP1. DFCP1 contains an unstructured N-terminal domain, a putative P-loop nucleotide-binding domain, a membrane binding domain and two FYVE domains. **b** Sequence alignments of the P-loop, switch I and switch II regions of DFCP1, GBP1, Atlastin1, Sey1P and KRAS. Boxes highlight residues critical for nucleotide hydrolysis in GBP1, Atlastin1 and KRAS. Asterisk indicates DFCP1 residues picked for mutational analysis. **c** DFCP1 binds ATP and ADP. DFCP1 WT (aa 1–410) was incubated with mantATP or mantADP (arrow), and nucleotide binding was measured by the incorporation of fluorescent mantATP/ADP. Curves are normalized to MBP background fluorescence. Bars: mean ± SD, 3 independent experiments. **d** DFCP1 does not bind guanosine nucleotides. Recombinant DFCP1 WT (aa 1–410) was incubated in the presence of mantGTP (4 independent experiments) or mantGDP (3 independent experiments) (arrow). Nucleotide binding was measured by incorporation of fluorescent mantGDP/GTP. Purified Cdc42 was used as a positive control for GTP binding (2 independent experiments). Curves are normalized to MBP (DFCP1) or GST (Cdc42) background fluorescence. **e** Increasing concentrations of DFCP1 WT (aa 1–410) were incubated with mantATP or mantADP, and the dissociation constant (Kd) was determined via nonlinear fitting: Kd ATP = 1068, (95% CI: 7.263–15.98), Kd ADP = 5434 (95% CI: 3138–9338), 3 independent experiments, bars: mean ± SD; nonlinear fit. **f** Measurement of phosphate release. Purified DFCP1 WT (aa 1–410) (10 μM) was incubated in the presence of ATP and the release of free phosphate was measured. Bars: mean ± SEM, 4 independent experiments, unpaired *t*-test, two-tailed *p*-value. **g** AlphaFold-generated model of full-length DFCP1. Shaded regions indicate functional domains. Candidates for amino acids

required for nucleotide binding (GKS, residues 192–194) are indicated in blue, and amino acids at the same position as the catalytic site of GBP1 (T189 in magenta, G190 in blue) and KRAS (G12, G13) are indicated. The nucleotide (GDP) and Mg²⁺ ion were extracted from GBP1 (PDB ID: 1f5n) after structural alignment. **h** DFCP1 binds ATP and identification of binding mutants. DFCP1 WT and mutants were incubated with mantATP (arrow), and nucleotide binding was measured by the incorporation of fluorescent mantATP. Curves are normalized to MBP background fluorescence. Bars: mean ± SD, 4 independent experiments. **i** Measurement of phosphate release by nucleotide-binding DFCP1 mutations. Purified DFCP1 and DFCP1 point mutants (aa 1–410) (10 μM) were incubated in the presence of ATP and the release of free phosphate was measured. Bars: mean ± SEM, 3 experiments, One-way ANOVA, Dunnett's multiple comparisons test, comparing against MBP. **j, i** Characterization of ATP-hydrolysis impaired DFCP1 point mutation. Purified DFCP1 WT and DFCP1 T189A (aa 1–410) were incubated in the presence of mantATP (arrow) and the binding of fluorescent mantATP was measured. Curves are normalized to MBP background fluorescence. Bars: mean ± SEM, 4 independent experiments. **j** DFCP1 T189A is unable to hydrolyze ATP. DFCP1 WT and DFCP1 T189A (aa 1–410) were incubated with ATP and the release of free phosphate was measured. Bars: mean ± SEM, 4 independent experiments, Ordinary One-way ANOVA, Dunnett's multiple comparisons test, comparing against MBP. **k** DFCP1 T189A has reduced ability to hydrolyze ATP. DFCP1 WT and DFCP1 T189A (aa 1-410) were incubated with ATP and the release of free phosphate was measured. Bars: mean ± SEM, 4 independent experiments, Ordinary One-way ANOVA, Dunnett's multiple comparisons test, comparing against MBP. Source data are provided as a Source data file.

performed size exclusion chromatography. Interestingly, whereas ADP-bound DFCP1 migrated as a monomer, ATPγS-loaded DFCP1 migrated as dimer (Fig. 2b, Supplementary Fig. S2b). As a control, we performed the same assay using DFCP1 K193A, the nucleotide-binding defective mutant we identified. This mutant migrated as a monomer and did not dimerize in the presence of ATPγS (Fig. 2b, Supplementary Fig. S2b), in line with our findings that dimerization of DFCP1 is ATP-dependent. Surprisingly, DFCP1 T189A, which bound ATP but was hydrolysis-deficient, also migrated as a monomer in the presence of ATPγS (Fig. 2b, Supplementary Fig. S2b). This suggests that efficient

ATP hydrolysis could require dimerization and that the T189 mutant could potentially break the dimer interface.

## DFCP1 ATP binding and hydrolysis are necessary for efficient omegasome constriction

We next addressed if nucleotide binding and hydrolysis are required for DFCP1 function during omegasome formation. To test whether cells expressing DFCP1 ATPase mutants show defects in omegasome formation, we developed knockout (KO)- and knockdown-based complementation systems (Supplementary Fig. S3). DFCP1 depleted

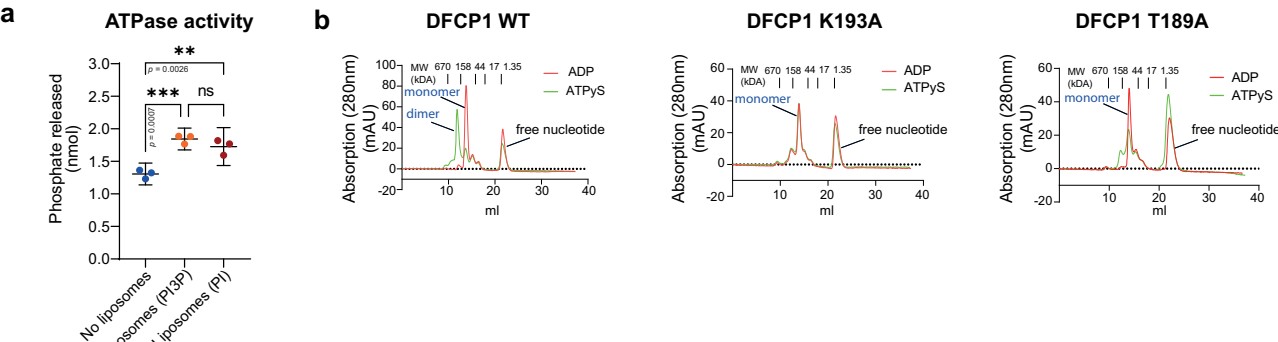

**Fig. 2 | DFCP1 forms dimers and its activity is stimulated by membranes.**
**a** DFCP1-FL was incubated in the absence or presence of Liposomes doped with either PtdIns3P (PI3P) or PI, and the release of free phosphate was measured. Bars: mean ± 95% CI, 3 independent experiments, Ordinary One-way ANOVA, Tukey's multiple comparisons test; ns: not significant. **b** Purified His6-MBP-DFCP1 WT or mutant (aa 1–410) were loaded with ADP or ATPγS and the apparent molecular weight was determined by size exclusion chromatography. Shown are the elution profiles in response to ADP or ATPγS loading. Relative molecular weights were determined by inclusion of a standard (Bio-Rad Gel filtration standard). The peak eluting at the higher molecular weight corresponds to a dimer, whereas the lower molecular weight corresponds to a monomer. The peak at the lower molecular weight represents free nucleotides. Both ADP and ATPγS-loaded DFCP1 K193A elute primarily as a single peak, corresponding to the molecular weight of a DFCP1 monomer. ADP-loaded DFCP1 T189A elutes as a complete monomer, and no side peaks corresponding to higher molecular weights are detectable. One representative experiment out of 3 independent experiments. Source data are provided as a Source data file.

cells were transduced with lentiviral vectors expressing low levels of either mNeonGreen (mNG)-tagged DFCP1 wild-type (WT) or the two ATP-binding or hydrolysis mutants K193A or T189A, in addition to mCherry- (mCh)-p62 or SNAP-LC3B as autophagosome markers. Structured illumination microscopy (SIM) revealed that mNG-DFCP1 WT localized to ring shaped omegasomes containing LC3B and p62 (Fig. 3a). To assess omegasome dynamics, the cells were starved with EBSS for 15 min and images were acquired with one frame taken every 2 s in starvation conditions (Fig. 3b, Supplementary Movie 1).

Autophagosome formation at omegasomes is a reproducible course of events[4,20]. Based on published data[4,20] and the tracking of DFCP1 WT omegasomes by live microscopy, we divided the omegasome formation process into three phases – expansion, maturation, and constriction (Fig. 3b). The expansion phase is characterized by a DFCP1 spot which is formed de novo, and to which p62 and LC3B are recruited a few seconds later[20,21] (Fig. 3b, Supplementary Fig. S4f). The spot grows and forms a ring-like structure, as previously observed[4,20]. We have defined the appearance of a ring with a visible lumen as the start for the maturation phase. LC3B and p62 are recruited to the initial DFCP1 spot and during the maturation, the DFCP1 ring grows, and LC3B and p62 span the lumen forming a disk-like structure (Supplementary Fig. S4c, g[20–22]). At the end of maturation, a p62 and LC3B-labeled phagophore extrudes out of the ring and forms a pocket (Supplementary Fig. S4c, g). This separation of the DFCP1-positive omegasome from the p62/LC3B-positive phagophore initiates the third phase – the constriction of the omegasome. The phagophore bends, buds out and is separated from the DFCP1-positive omegasome, thereby forming the nascent autophagosome. During this process, the shrinking omegasome remains connected to the growing autophagosome, which is positive for both p62 and LC3B, but also weakly for DFCP1 (Fig. 3a, b; Supplementary Fig. S4g). Finally, the weak DFCP1 signal leaves the autophagosome, and DFCP1 at the collapsed omegasome disappears, leaving a p62/LC3B-positive-finished autophagosome (Fig. 3b, Supplementary Fig. S4g).

We next analyzed the dynamic formation of omegasomes in DFCP1 mutant cells. Whereas these omegasomes appeared morphologically similar to the WT omegasomes (Supplementary Fig. S4c, g), we asked if they might have difficulties in recruiting LC3B or p62. To determine how many omegasomes successfully recruit p62, we tracked forming omegasomes and then measured p62 or LC3B intensity at the omegasome. We found that p62 was recruited at similar rates in cells expressing either WT and ATPase mutant DFCP1 (Supplementary Fig. S4d). We found that on average, 80% of all newly formed omegasomes acquired p62, whereas approximately 20% of omegasomes did not acquire p62 (Supplementary Fig. S4e). Of note, there was no difference in p62 recruitment between cells expressing WT or ATPase mutant DFCP1 (Supplementary Fig. S4e). LC3B recruitment to omegasomes behaved similarly. Approximately 75% of omegasomes recruited LC3B, whereas ~25% failed to recruit LC3B (Supplementary Fig. S4i). Also here, we did not observe any difference between cells expressing WT or ATPase mutant DFCP1. Thus, the initial formation of an omegasome and recruitment of p62 and LC3B occurs independently of DFCP1 ATPase activity.

Next, we measured the total lifetime of an omegasome, from the initial appearance of a DFCP1 dot to the complete disappearance. Detailed analysis and tracking revealed a significant increase in the omegasome lifetime in cells expressing mutant DFCP1 (Fig. 3g, Supplementary Fig. S4e, Supplementary Movie 2). While DFCP1 WT used on average 330 s to form omegasomes as previously described[21,4,20], the ATP-binding mutant DFCP1 K193A used 500 s, whereas the ATPase-defective mutant DFCP1 T189A needed 450 s to complete the process (Fig. 3d). To understand which step in omegasome biogenesis was affected in the mutants, we measured the duration of the three individual phases. Omegasomes in cells expressing DFCP1 WT used approximately 100 s for each of the two first phases, whereas the last phase took 150 s. Surprisingly, cells expressing either of the two DFCP1 mutants showed a nearly unchanged duration for the first two phases. In contrast, the constriction phase was markedly delayed compared to WT (Fig. 3e). We confirmed these findings also using DFCP1 knockdown cell lines, which were rescued with stable expression of low levels of siRNA-resistant WT or mutant mNG-DFCP1 in combination with SNAP-LC3B (Supplementary Fig. S3f–i, Supplementary Fig. S4h, j, Supplementary Movie 3, Supplementary Movie 4).

To validate our findings with quantitative data, we repeated the omegasome tracking using the KO rescue cell lines and automatically measured the size of the omegasome over time. Cells expressing either WT or mutant DFCP1 showed very similar dynamics during the initial phases of omegasome formation. We observed a slight delay in the expansion of omegasomes from start to maximum diameter with the ATP-binding defective K193A mutant (Fig. 3f). After reaching the maximum diameter, omegasomes expressing WT DFCP1 rapidly and completely constricted. In contrast, omegasome constriction was

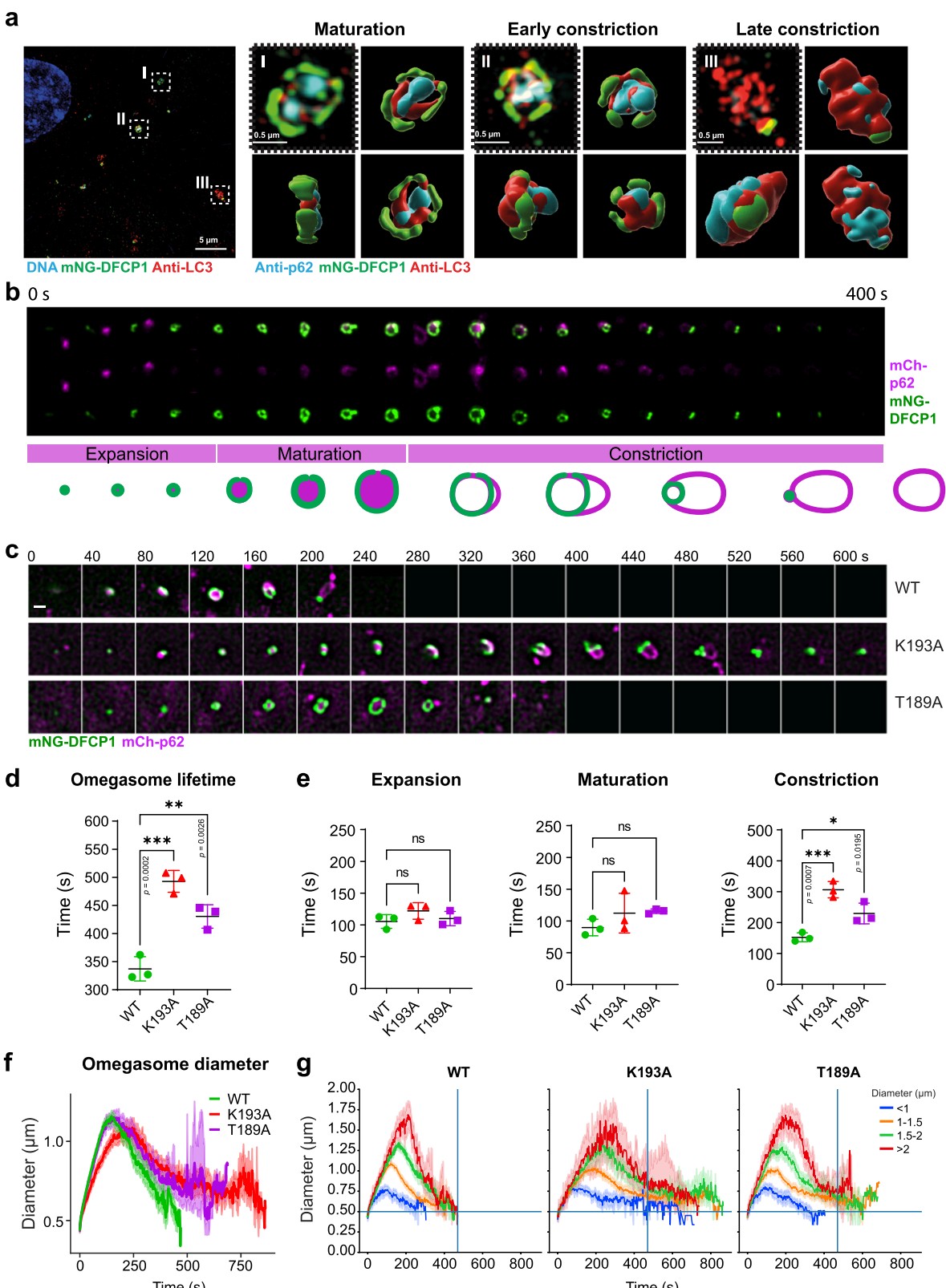

markedly delayed in cells expressing either the hydrolysis-defective or nucleotide-binding defective mutants (Fig. 3f). These omegasomes did not completely constrict but instead stalled at ~0.8 μm diameter for prolonged periods of time.

To further analyze this phenotype, we classified individual omegasome tracks by the maximum omegasome diameter (Fig. 3g) or omegasome lifetime (Supplementary Fig. 4b). This analysis showed

that in cells expressing WT DFCP1, omegasome expansion and constriction were highly ordered and showed similar dynamics independently of the final size of the omegasome. Consequently, the lifetimes of omegasomes with WT DFCP1 fell within a small range, between 200 and 450 s (Supplementary Fig. S4a). In contrast, expansion and constriction of omegasomes in cells expressing either DFCP1 mutant showed ordered dynamics. While all omegasomes – independently of

**Fig. 3 | DFCP1 mutants show delayed omegasome constriction. a** 3D reconstruction of omegasomes in maturation and constriction phases. U2OS DFCP1 KO lines stably expressing mNG-DFCP1 WT were starved in EBSS, fixed, stained for p62, LC3B and mNG and SIM imaging was performed. Shown are examples from three different stages of omegasome formation, and their reconstruction from different angles. Scale bar: 5 μm, insets: 0.5 μm. **b** U2OS DFCP1 KO lines stably expressing mNG-DFCP1 WT and mCh-p62. Cells were starved in EBSS and SoRa live-cell super-resolution imaging was used to capture omegasome formation. Omegasome formation was divided into three phases as indicated. The scheme illustrates how DFCP1 and p62 co-localize throughout an omegasomes' life cycle. Quantification of events in (**d**–**g**). Scale bar: 0.5 μm. **c** U2OS DFCP1 KO lines stably expressing mNG-DFCP1 WT or -mutant and mCh-p62. Cells were starved with EBSS and imaged live to capture omegasome formation. Shown are representative stills of omegasomes analyzed in (**d**–**g**). Scale bar 0.5 μm. **d** Quantification of the total omegasome lifetime, with a DFCP1 spot as beginning and end point. Each plotted point represents the mean value of one experiment. Bars: mean ± SD, 3 independent experiments, analyzing in total 58 (WT), 16 (K193A) and 49 (T189A) omegasomes. One-way ANOVA, Dunnett's post hoc test, comparing against WT. **e** Duration of the individual phases of omegasome formation, dataset from (**d**). Each plotted point represents the mean value of one experiment. Bars: mean ± SD, 3 experiments, Ordinary One-way ANOVA, Dunnett's multiple comparisons test, comparing against WT. 3 independent experiments; ns: not significant. Analyzed omegasomes: Initiation: WT (76), K193A (54), T189A (81); Maturation: WT (99), K193A (81), T189A (117); Constriction: WT (78), K193A (47), T189A (99). **f** Same dataset as in (**d**). The signal of DFCP1 was segmented, and mNG-DFCP1 diameter was measured over time. 3 independent experiments, total number of tracks: WT (329), K193A (133), T189A (266). Mean ± 95% CI. **g** Same dataset as in (**f**). Individual omegasomes have been classified by their maximum diameter and their diameter was plotted over time. The vertical line indicates complete disappearance of WT omegasomes, the horizontal line indicates successful constriction (to a diameter smaller than 0.5 μm). Shown is the mean, bands indicate ± 95% CI. Source data are provided as a Source data file.

their size – showed a similar short initial constriction phase, the final constriction with DFCP1 mutants was slower than with WT DFCP1 and the majority of omegasomes stalled once they reached ~0.8 μm in diameter (Supplementary Fig. S4b).

The phenotype of cells expressing the ATPase-defective DFCP1 (T189A) was milder than with the K191A mutation. These cells still showed an increase in omegasome lifetime and delays in closing large omegasomes, but omegasome closure occurred faster than with the K193A mutant. As this mutation retained ~20% residual ATPase activity (Fig. 1k), it might also partially retain its biological function.

Based on these data, we conclude that the ATP-binding and hydrolysis activities of DFCP1 are necessary to efficiently complete the constriction of at least a subset of omegasomes.

## DFCP1 ATPase mutants cause increased numbers of omegasomes

Based on our finding that DFCP1 ATPase is required for omegasome constriction, we asked if this defect has consequences for autophagy. Upon starvation with EBSS, cells expressing the ATP-binding or ATPase-defective mutants showed an increased number of DFCP1 puncta compared to DFCP1 WT, and both mutants had more DFCP1 dots and rings (Fig. 4a, b).

The increased number of omegasomes in the DFCP1 mutants can be either a consequence of increased omegasome formation or prolonged persistence. To investigate whether omegasome formation was increased, we made use of another omegasome marker, WIPI2[9,10]. WIPI2 is recruited by PtdIns3P to phagophores originating at omegasomes. It appears nearly at the same time as DFCP1 but dissociates earlier[9] (Supplementary Fig. S5a). Thus, WIPI2 represents a good marker for omegasome formation independently of DFCP1. We analyzed endogenous WIPI2 puncta 2 h after EBSS treatment in DFCP1 WT and mutants. As expected, a portion of the WIPI2 dots co-localized with DFCP1-positive omegasomes, likely representing early stages of autophagosome formation (Supplementary Fig. S5b). Interestingly, although the number of DFCP1 puncta was higher in cells expressing the mutants, neither the number of WIPI2 puncta nor the portion of DFCP1-positive WIPI2 puncta was changed (Fig. 4c–f), suggesting that omegasome initiation was not increased. On the other hand, the portion of DFCP1 puncta positive for WIPI2 was reduced in cells expressing the mutants, indicating that the DFCP1-positive omegasomes that accumulated in the mutant cells likely represent a later, WIPI2 negative, stage of omegasomes.

We conclude that whereas the number of omegasomes increases in the DFCP1 mutant cells, the onset of omegasome formation is not affected. This is in line with our findings that the initial dynamics of omegasome formation is not altered in DFCP1 mutant cells.

## Autophagic flux of p62 is compromised in DFCP1 ATPase mutants

It has remained a paradox that depletion of DFCP1 by siRNA does not have any effect on the flux of LC3B[4]. To address whether LC3B flux is affected upon specific modulation of DFCP1 ATPase activity, we measured the sum intensity of endogenous LC3B puncta in DFCP1 WT and DFCP1 mutant rescue cells by automated analysis of confocal micrographs. As expected, numerous LC3B puncta were detected upon EBSS starvation in DFCP1 WT cells (Supplementary Fig. S6a). DFCP1 ATP-binding deficient K193A mutant cells showed a small, but significant increase in the sum intensity of LC3B dots, whereas cells expressing the ATP-locked T189A mutant only showed a tendency to accumulate LC3B as compared to DFCP1 WT (Fig. 4g, Supplementary Fig. S6a).

It has been reported that p62 localizes to phagophores that are formed at omegasomes, and that this recruitment occurs independently of LC3B[23]. In line with this, we noticed that while LC3B was present on many other structures in addition to omegasomes, the majority of p62 puncta localized with DFCP1-positive omegasomes (Supplementary Fig. S6a). Importantly, when we measured the sum intensity of endogenous p62 puncta, both DFCP1 ATPase mutant cell lines showed a more than twofold increase in p62 dot intensity as compared to DFCP1 WT cells upon EBSS treatment (Fig. 4h). Moreover, in cells expressing high amounts of DFCP1, p62 hyper-accumulated inside DFCP1 ATPase deficient omegasomes (Supplementary Fig. S6b).

To address whether the increased level of p62 dots observed in the mutants could be explained by an impaired autophagic flux, we measured the sum intensity of p62 dots per cell in fed or EBSS starved cells in the presence or absence of Bafilomycin A1, which inhibits lysosomal degradation activity. This analysis showed a nearly threefold increase in p62 dot sum intensity in the EBSS-treated mutant cells (Fig. 4i), thus confirming our previous results. In addition, we observed a similar effect in fed cells, indicating that basal autophagy was also impaired in DFCP1 depleted or ATPase deficient cells (Fig. 4j, k, Supplementary Fig. S7a–c). Addition of Bafilomycin A1 indeed prevented the lysosomal degradation of p62, as the sum intensity of p62 dots clearly increased (Fig. 4i). Importantly, when lysosomal degradation was blocked with Bafilomycin A1, there was no difference in the p62 levels between DFCP1 WT or ATPase-defective cells (Fig. 4i). This indicates that p62 accumulates in the DFCP1 ATP-binding and ATPase-defective mutants due to a delayed autophagic flux, rather than increased onset of autophagy. This is also consistent with our finding that the number of WIPI2 puncta were unaffected. Taken together, our data show that DFCP1 is required for efficient autophagic flux of p62, and that this depends on its ability to bind and hydrolyze ATP.

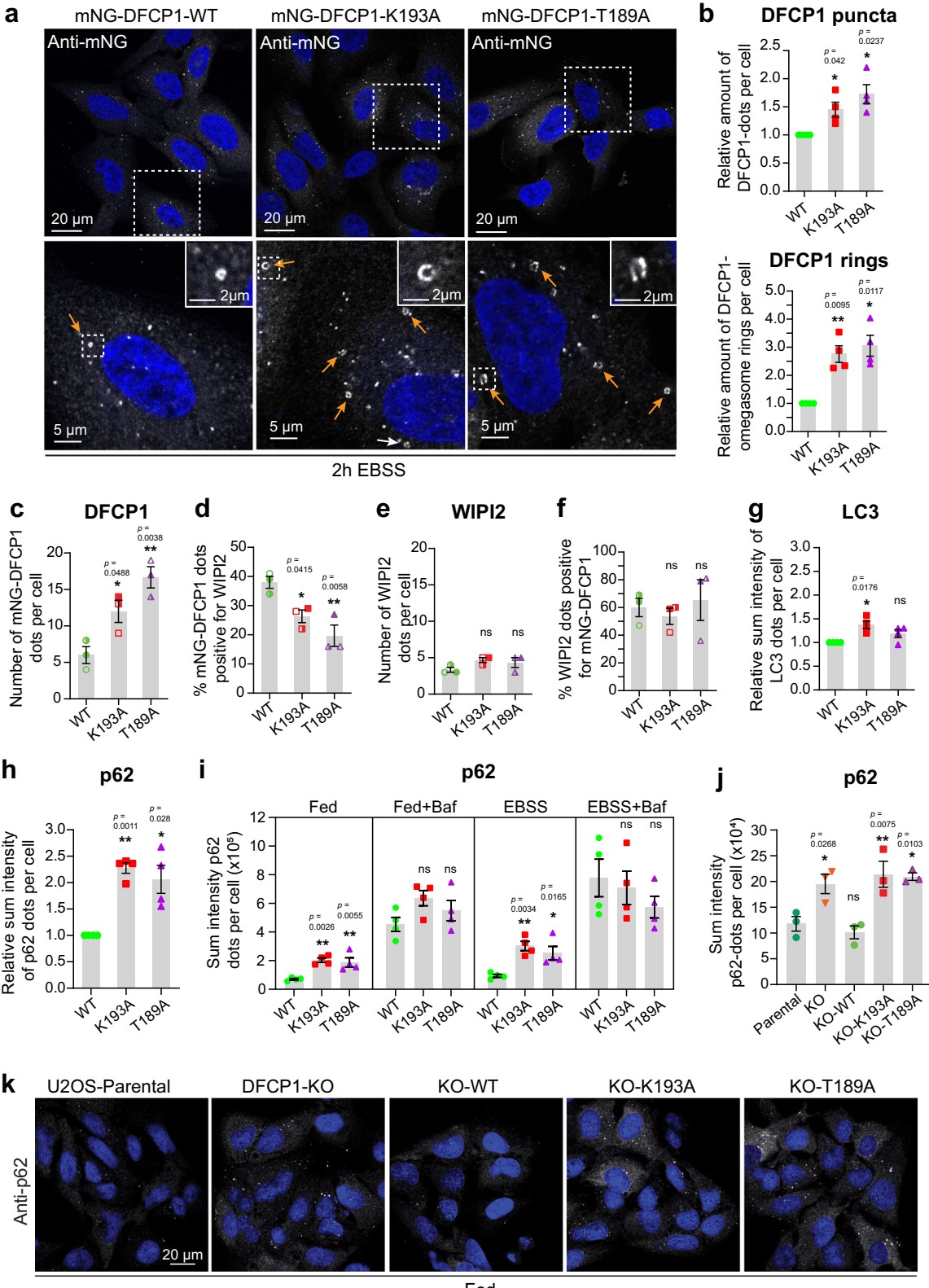

## DFCP1 ATPase is required for selective autophagy

Despite the prominent localization at pre-autophagic omegasomes and its wide use as autophagy marker, DFCP1 does not show strong effects on LC3B lipidation[4]. Also in our system, we did not find any effects of DFCP1 KO or DFCP1 ATPase mutants on LC3B lipidation (Fig. 4g, Supplementary Fig. S6a). To test if loss or mutation of DFCP1 affects bulk autophagy, we measured sequestration of bulk cargos by

flow cytometry using a cytoplasmic mKeima reporter[24,25] in U2OS and RPE-1 cells (Fig. 5a, Supplementary Fig. S8a–c). These assays showed no effect of DFCP1 KO or DFCP1 ATPase mutants on sequestration of the bulk reporter.

However, since we observed that p62 accumulates in DFCP1 ATPase mutants in both fed and starved conditions, we asked if the DFCP1 ATPase mutants are impaired in selective autophagy. p62 has

**Fig. 4 | p62 accumulates in DFCP1 mutants defective in ATP binding or hydrolysis. a–i** U2OS cells stably expressing siRNA-resistant mNG-tagged DFCP1 alleles (WT, K193A or T189A) were depleted of endogenous DFCP1 by siRNA transfection for 2 d and starved in EBSS for 2 h. **a** Fixed cell confocal imaging showing accumulation of different omegasome stages in DFCP1 WT and mutant cells. Cells were stained with anti-mNG antibody. Arrows indicate DFCP1-positive omegasome rings. Representative image for quantifications in (**b**). **b** Number of mNG-DFCP1 dots (upper) and DFCP1 rings (lower) per cell. Error bars: mean ± SEM, 4 independent experiments. Each point represents the mean value of one experiment. One-sample *t*-test, two-sided. Analyzed cells: WT (643), K193A (650), T189A (632). **c–f** Quantification of DFCP1 and WIPI2 puncta. Cells were treated as described above, fixed, stained with anti-WIPI2 antibody and analyzed by confocal microscopy (representative images in Supplementary Fig. S5b). Number of mNG-DFCP1- (**c**) or WIPI2- (**e**) positive structures were scored automatically. Co-occurrence between mNG-DFCP1 and WIPI2 in (**d**) and (**e**). Error bars: mean ± SEM, 3 independent experiments. Each point represents the mean value of one experiment indicated by different colored or transparent objects. Ordinary one-way ANOVA, Dunnett's post hoc test. Analyzed cells: WT (268), K193A (273), T189A (227). **g** Quantification of LC3B dots. Cells were treated as described above, fixed, stained with anti-LC3B and analyzed by confocal microscopy. Shown is the relative sum intensity of LC3B dots per cell. Error bars: mean ± SEM, 4 independent experiments. Each point represents the mean value of one experiment. One-sample *t*-test, two-

sided. Analyzed cells: WT (458), K193A (461), T189A (446). **h** Quantification of p62 dots. Cells were treated as described as above, fixed, stained with anti-p62 and analyzed by confocal microscopy. Shown is the relative sum intensity of p62 dots per cell. Error bars: mean ± SEM, 4 experiments. Each point represents the mean value of one experiment. One-sample *t*-test, two-sided. Analyzed cells: WT (643), K193A (650), T189A (632). Same dataset as in (**b**). **i** Cells were treated as described above and cultured either in complete medium or in EBSS for 2 h, in the presence or absence of 100 nM Bafilomycin A1 to assess autophagic flux. Cells were fixed, stained with anti-p62 antibody, and analyzed by confocal microscopy. Shown is the sum intensity of p62 dots per cell. Error bars: mean ± SEM, 4 independent experiments. Each point represents the mean value of one experiment. One-way ANOVA, Dunnett's multiple comparisons test comparing against WT within each treatment group. In total >400 cells were analyzed per cell line per condition. **j, k** Parental U2OS, DFCP1 KO and KO rescue lines stably expressing mNG-DFCP1 WT, K193A or T189A were grown in complete medium. **j** Quantification of p62 dots. Cells were fixed and stained with anti-p62 antibody and Hoechst 33342 and analyzed by confocal microscopy. Shown is the quantification of the sum intensity of p62 dots per cell. Error bars: mean ± SEM, 3 independent experiments. Each point represents the mean value of one experiment. Ordinary one-way ANOVA, Dunnett's post hoc test. Cells analyzed: parental (788); KO (703); WT (732), K193A (779), T189A (766). **k** Representative images for cells analyzed in (**j**). ns not significant. Source data are provided as a Source data file.

been implicated in several types of selective autophagy, such as aggrephagy and mitophagy[26–28].

To induce aggrephagy, we treated cells stably expressing mCh-DFCP1 with Puromycin. Puromycin treatment leads to the formation of large p62-containing aggregates, which are heavily ubiquitinated and cleared by autophagy[29]. DFCP1 localized to a subset of these aggregates, forming characteristic omegasomes positive for LC3B (Fig. 5b, Supplementary Fig. S9a, b). Strikingly, DFCP1 KO cells had more than a twofold increase in p62 aggregates upon Puromycin treatment compared to the WT (Fig. 5c). This could be rescued by WT DFCP1, but not the ATP-binding and ATPase mutants (Fig. 5d, Supplementary Fig. S9c). Thus, DFCP1 and its ATPase activity are necessary for efficient aggrephagy.

To investigate if DFCP1 is necessary for autophagic degradation of mitochondria, we used the iron chelator Deferiprone (DFP) to induce mitophagy[30] in cells stably expressing both mNG-DFCP1 and a mitochondrial marker (mitoSNAP). Using live-cell imaging, we observed that DFCP1 formed omegasomes at the surface of mitochondria, and that mitochondrial fragments were channeled through the omegasome ring (Fig. 5e, Supplementary Fig. S10a, Supplementary Movie 5, Supplementary Movie 6). Importantly, the mitochondrial fragments were surrounded by the weakly DFCP1-positive phagophore, indicating that it had been engulfed by the autophagosome (insets Fig. 5e, Supplementary Fig. S10a). Following engulfment and constriction of the omegasome, autophagosomes were released from the mitochondria (Supplementary Fig. S10a).

The importance of DFCP1 for mitophagy was quantified using RPE-1 cells stably expressing a mitochondrial mKeima probe, mito-mKeima[24,31]. When mito-mKeima is transported to acidic lysosomes for degradation, the excitation maximum changes and mitophagy can be determined by flow cytometry as an increased ratio of lysosomal mito-mKeima. While control cells displayed a nearly twofold increase in lysosomal mito-mKeima when mitophagy was induced, DFCP1 depleted cells had a strongly reduced capacity to deliver mito-mKeima to lysosomes (Fig. 5f, Supplementary Fig. S10b). This indicates that DFCP1 is required for proper DFP-induced mitophagy.

To address whether DFCP1 ATPase activity is necessary for mitophagy, we stably expressed DFCP1 WT or mutants in the mito-mKeima RPE-1 cells depleted of endogenous DFCP1 and performed mitophagy assays (Fig. 5g, Supplementary Fig. S10c, d). Both ATP-binding and ATPase-defective mutants of DFCP1 were impaired in mitophagy (Fig. 5g, Supplementary Fig. S10c, d), indicating that the ATPase activity of DFCP1 is necessary for efficient mitophagy.

When examining the phenotype of U2OS DFCP1 KO cells, we noticed that they contained an increased number of micronuclei. Image quantifications showed that DFCP1 KO and ATPase deficient cells had a more than twofold increase in the number of micronuclei per cell as compared to parental or DFCP1 WT cells (Fig. 5i, Supplementary Fig. S9d). This phenotype was also confirmed in A431 DFCP1 KO cells (Supplementary Fig. S9g, h). Similarly, U2OS cells acutely depleted for DFCP1 showed an increase in the number of micronuclei per cell (Supplementary Fig. S9e, f). Importantly, we noticed that DFCP1 localized to a subset of micronuclei, colocalizing with p62. Super-resolution microscopy revealed that endogenously GFP-tagged DFCP1 formed characteristic omegasome rings at the surface of micronuclei together with p62 (Fig. 5h). These results suggest that DFCP1 promotes autophagic degradation of micronuclei, in addition to mitochondria and protein aggregates. We conclude that DFCP1 is specifically required for selective autophagy.

## Discussion

DFCP1 translocates to omegasomes in a PtdIns3P dependent manner following amino acid starvation[4]. Although DFCP1 has been widely used as an omegasome marker, its molecular function is so far completely unknown. Here, we have found that DFCP1 is a large ATPase, and its ATPase activity mediates the timely constriction of omegasomes in selective autophagy.

Our bioinformatic analyses showed that DFCP1 bears highest similarity to large GTPases such as Dynamin, GBP1 and Atlastins (Fig. 1b). In contrast to these GTP-binding proteins, we unexpectedly found that DFCP1 binds ATP and functions as an ATPase. A similarly altered nucleotide specificity has been found for EHD and related proteins, which are related to Dynamin, but are selective for ATP instead of GTP[15].

Structurally, ATP binding by DFCP1 is highly similar to other NTPases. The P-loop GKS motif, which is highly conserved in most nucleotide-binding proteins, plays a critical role for nucleotide binding. However, the mechanism of nucleotide hydrolysis is less clear and large Dynamin-related NTPases have evolved several different mechanisms. In DFCP1, we were able to identify one important residue, Threonine-189, which we found to be critical for ATP hydrolysis. A corresponding amino acid can also be found in the GTPase Irga6 (T78). In Irga6, residues in switch I can contact the ribose of the nucleotide bound in the opposing protein, which stabilizes dimerization[32]. This dimerization promotes GTP hydrolysis. Importantly, a mutation in Threonine-78 of Irga6[33], which corresponds to Threonine-189 in

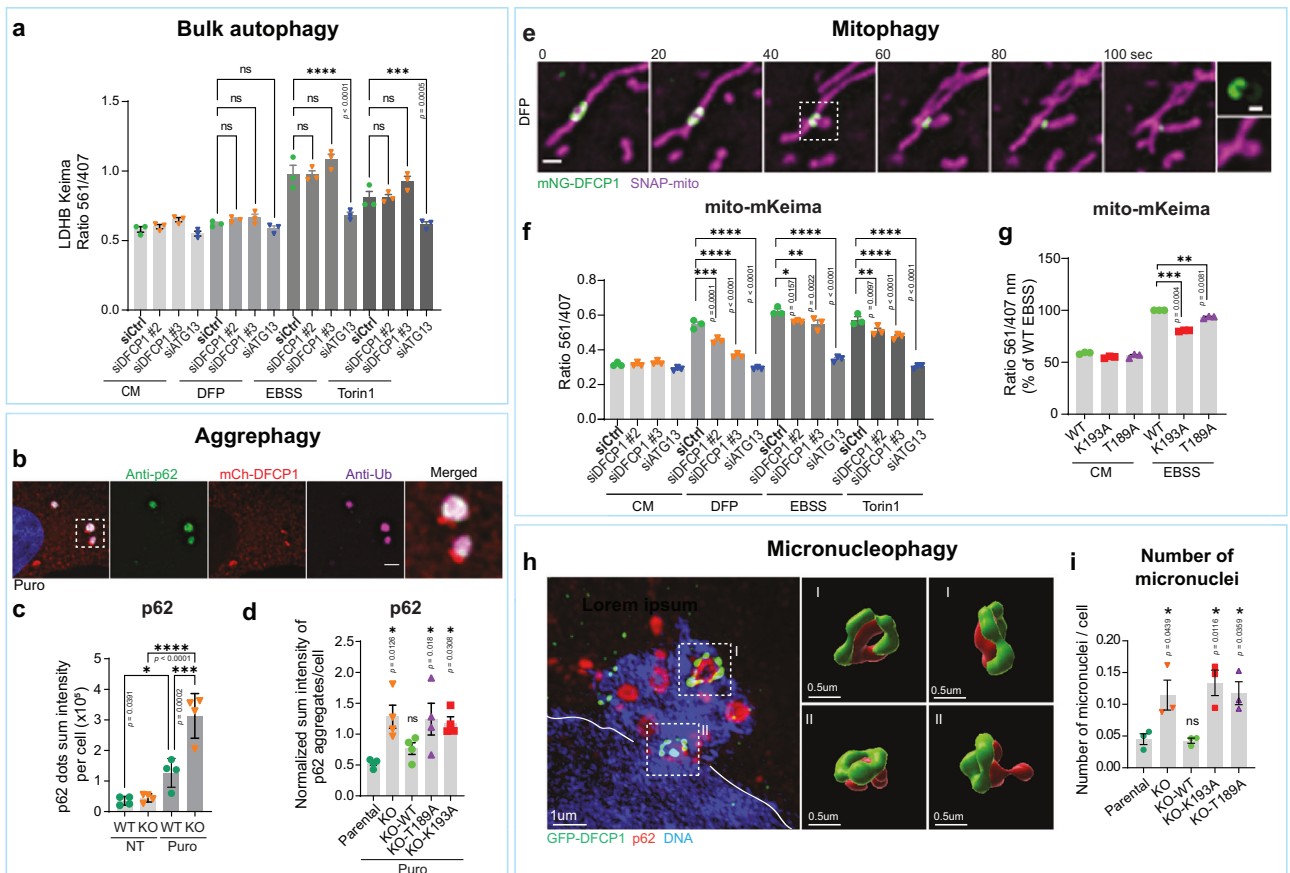

**Fig. 5 | DFCP1 ATPase regulates selective types of autophagy. a** RPE-1 cells induced to express LDHB-mKeima were transfected with control-, DFCP1- or ATG13 siRNA for 2 d. Cells were cultured for 16 h either in complete medium, in 0.5 mM DFP, in EBSS, or treated with 50 nM Torin1. Cells were harvested and subjected to Flow Cytometry to determine the ratio of lysosome-localized LDHB-mKeima to free, cytoplasmic LDHB-mKeima. Bars: ± SEM, 3 independent experiments, One-way ANOVA, Sidak's multiple comparisons test. **b** U2OS cells stably expressing mCh-DFCP1 were treated with 10 µg/ml Puromycin for 3 h, fixed, and stained with antibodies against p62 and ubiquitin. Representative image for 20 cells, scale bar: 2 µm. **c** U2OS WT and DFCP1 KO cells were treated or not with 2.5 µg/ml Puromycin for 3 h, fixed, stained with anti-p62 antibody, and analyzed by confocal microscopy. Graphs represent quantification of the relative sum intensity of p62 dots per cell. One-way ANOVA, Sidak's multiple comparisons test. Bars: mean ± SD, 4 independent experiments. Each plotted point represents the mean value of one experiment. Cells analyzed: WT-NT (1175), KO-NT (1042), WT-Puro (1465), KO-Puro (1381). **d** Parental U2OS, DFCP1 KO and KO rescue lines stably expressing mNG-DFCP1 WT, K193A or T189A were treated with 2.5 µg/ml Puromycin for 2 h, fixed, stained with anti-p62 antibody, and analyzed by confocal microscopy. Graphs represent quantification of the relative sum intensity of p62 aggregates per cell. Values have been normalized by the mean of the mean. Bars: mean ± SEM, 4 independent experiments, One-way ANOVA, Dunnett's post hoc test, comparing against U2OS parental. Cells analyzed: parental (1732), KO (1716), KO-WT (2392), KO-T189A (2601), KO-K193A (2263). **e** Example of a mitophagy event. U2OS cells stably expressing mNG-DFCP1 WT and SNAP-mito were treated for 18 h with 0.5 mM DFP and imaged live. The inset shows a piecemeal of mitochondria, which is engulfed by the phagophore. Note that the phagophore is weakly positive for mNG-

DFCP1 (inset). Representative movie for >10 events. Scale bar: 1 µm, inset 0.5 µM. **f** RPE-1 cells induced to express a mitochondrial mKeima probe were depleted for DFCP1 for 2 d and were cultured for 16 h in either complete medium (CM), 0.5 mM DFP, EBSS or 50 nM Torin1, harvested and analyzed by Flow Cytometry to determine the ratio of lysosome-localized mito-mKeima to free, cytoplasmic mito-mKeima. Bars: ± SEM, 3 independent experiments, One-way ANOVA, Sidak's multiple comparisons test. **g** RPE-1 cells stably expressing mitochondrial mKeima probe and siRNA-resistant mNG-DFCP1-WT, K193A, or T189A were depleted of endogenous DFCP1 by siRNA transfection for 2 d in complete medium (CM) before starvation for 16 h in EBSS to induce mitophagy. Cells were harvested and analyzed by Flow Cytometry to determine the ratio of lysosome-localized mito-mKeima to free, cytoplasmic mito-mKeima. Values were normalized to EBSS-treated WT. Bars: ± SEM, 3 independent experiments, One-sample t-test, two-sided. **h** GFP-DFCP1 knockin cells have been fixed and stained with p62 and GFP-antibodies and Hoechst 33342 and analyzed by SIM-microscopy. Shown is a micronucleus decorated with omegasome rings, which have been 3D-rendered and are shown from different angles. The border of the main nucleus is indicated by a white line. **i** U2OS parental cells, DFCP1 KO and KO rescue lines stably expressing mNG-DFCP1 WT, K193A or T189A were grown in complete medium, fixed, stained with Hoechst 33342, analyzed by confocal microscopy and the number of micronuclei determined. Error bars: mean ± SEM, 3 independent experiments. Each plotted point represents the mean value of one experiment. One-way ANOVA, Dunnett's post hoc test. Cells analyzed: parental (788); KO (703); WT (732), K193A (779), T189A (766). Same dataset used to quantify p62 intensity in Fig. 4j, k. Images of micronuclei in Supplementary Fig. S9d. ns not significant. Source data are provided as a Source data file.

DFCP1, interferes with dimerization and strongly reduces nucleotide hydrolysis.

We found that wild-type but not ATPase-defective (T189A) DFCP1 forms an ATP-dependent dimer. This suggests that dimerization of DFCP1 could be required for effective nucleotide hydrolysis and that DFCP1 employs a similar mechanism as Irga6 for nucleotide hydrolysis. Most large NTPases of the Dynamin family bind to membranes, and in many cases, this binding can stimulate the nuclease activity. We found

that DFCP1 ATPase activity is stimulated by the presence of membranes. The observed increase in hydrolysis was moderate (-1.4 fold) but robust, suggesting that – similar to other NTPases – binding of DFCP1 to membranes could be an important regulator of ATPase activity. So far, it is not clear if this is due to a conformational change in response to membrane binding which stimulates nucleotide hydrolysis, or if clustering of DFCP1 on membranes promotes dimerization and thereby enhances ATP hydrolysis.

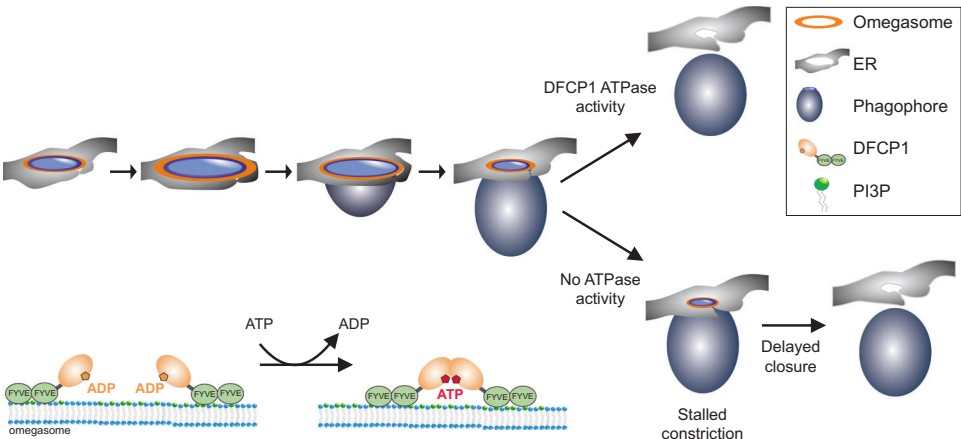

**Fig. 6 | Model of how DFCP1 acts at the omegasome during selective autophagy.** Upon induction of autophagy, PtdIns3P is generated at specialized subregions of the ER, at omegasomes. A DFCP1 spot is formed de novo, to which p62 and LC3B are recruited a few seconds later. The spot grows and forms a ring-like structure. Omegasomes first expand, and the phagophore, likely originating from a cytoplasmic seed vesicle, extrudes out of the ring and forms a pocket. Then, the omegasome ring constricts and finally, the phagophore is closed and becomes an autophagosome. DFCP1 localizes to the omegasome throughout the whole process of autophagosome formation. ATPase activity of DFCP1 is required during the constriction of the omegasome. In cells expressing WT DFCP1, the omegasome rapidly constricts. In contrast, in cells expressing ATPase mutants of DFCP1, the constriction stalls, delaying the closure of the omegasome. On a molecular scale, ATP binding and hydrolysis controls dimerization state of DFCP1 which could provide the mechanochemical energy to drive this constriction.

Our functional data indicate that the DFCP1 ATPase is involved in the maturation of omegasomes, specifically during omegasome constriction. It is worth noting that expansion, maturation and initial constriction of omegasomes were largely unaffected by mutations in the DFCP1 ATPase domain, indicating that these steps might not rely on DFCP1 function, and DFCP1 could only localize as a passenger to the PtdIns3P-rich omegasomes. Only the final step of constriction was strongly delayed in DFCP1 ATPase mutants, suggesting that this step relies on ATPase activity.

This delay of the last constriction step in DFCP1 mutants bears a striking similarity to the constriction of membrane necks by Dynamin and the stalled constriction of both hydrolysis-defective and nucleotide-binding Dynamin mutants[34]. Dynamin mutants accumulate stalled membrane necks which are defective in the final scission steps. Dynamin binds to membranes by its C-terminal PH domain and forms spiral-like assemblies at membrane necks. Cycles of nucleotide loading and hydrolysis cause these assemblies to constrict, which ultimately drives membrane scission[35]. Similar to Dynamin, DFCP1 can also bind to membranes by its two C-terminal FYVE domains, which contribute to anchor the protein to the membrane. It is tempting to speculate that the nucleotide hydrolysis cycle of DFCP1 could lead to repeated dimerization and dissociation of the ATP-binding domain, which would lead to a stepwise constriction of the omegasome (Fig. 6). While the observed phenotypes support such a model, the structural basis is less clear. Moreover, most Dynamin-related proteins have a bundle of antiparallel alpha helices – the so-called stalk region - adjacent to the nucleotide-binding G domain. This stalk region acts as dimerization interface and allows the assembly of large multimers, which are critical to drive constriction of membranes. We noted that Alphafold-generated predictions of DFCP1 do not show the typical antiparallel stalk region directly adjacent to the G domain. However, our predictions show an antiparallel alpha-helix which is formed between the two FYVE domains, which could be a derivative of the stalk domains in other dynamin-related proteins and have similar functions. Thus, it is not clear if and how DFCP1 can indeed generate mechanical forces to drive membrane constriction. An alternative model could be that DFCP1 acts as a tether between membranes, similar to Atlastins.

Since p62 positive forming autophagosomes were connected to the DFCP1 rings for the whole duration of their constriction period, we speculate that ring constriction is an important prerequisite for sealing of the extended phagophore, and to release the nascent autophagosome. Both the ATP-binding defective mutant and the ATP-locked mutant showed a similar phenotype, which can probably be attributed to our finding that both mutants are unable to dimerize. Functionally, the slower constriction of the omegasome translates to a delay in autophagic flux of p62.

Previously it had been concluded that depletion of DFCP1 has no effect on autophagic flux, measured by LC3B following amino acid starvation[4]. Our experiments, using a free cytoplasmic Keima probe, confirmed that DFCP1 does not play a role in bulk autophagy. This finding is in line with the fact that DFCP1 – despite being in an ancient protein - does not exist in simple eukaryotes and has been lost in several metazoan species, e.g., *Drosophila melanogaster*. These organisms have nevertheless well-functioning autophagic pathways, indicating that DFCP1 is unlikely to be part of the core autophagy machinery and is dispensable for bulk autophagic uptake.

This raises the question which cellular functions DFCP1 fulfils. While we did not observe any defects in the sequestration of bulk cargos, we found a striking dependence on DFCP1 in selective autophagy. This was unanticipated given the fact that DFCP1-containing omegasomes are typically induced by amino acid starvation. However, such starvation does not only induce bulk autophagy but also accumulation of various cargos for selective autophagy[30,36], and this could be the explanation why omegasomes are observed after starvation. In addition to p62-dependent aggrephagy, we also found that efficient degradation of mitochondria and micronuclei depends on the ability of DFCP1 to bind and hydrolyze ATP. Importantly, we did not observe any differences in omegasome morphology for the different types of selective autophagy. Thus, omegasome morphology, as well as the progression of omegasome formation and constriction appear to be very similar, independently of the cargo.

Further studies are needed to establish the exact functions of DFCP1 in mitophagy and micronucleophagy, but it is striking that several types of selective autophagy depend on DFCP1. While it is interesting that KO of DFCP1 results in accumulation of micronuclei, we cannot exclude the possibility that this increase is a secondary effect on cell division caused by impaired selective autophagy[37,38]. However, the presence of multiple DFCP1-labeled omegasomes on micronuclei supports a direct role for DFCP1 in micronucleophagy.

Given the presence of DFCP1 at starvation-induced phagophores, how can we explain the preferential requirement of this ATPase for selective over bulk autophagy? One possibility is that DFCP1 only

becomes essential when phagophores reach a certain size, as typical for selective autophagy, and that other ER remodeling factors are sufficient to sustain formation of smaller autophagosomes that mediate bulk autophagy. All selective autophagic pathways which require DFCP1 have in common that the sequestered cargos are large (100 s of nm to several μm). This consequently requires large omegasome structures and the extrusion of large isolation membranes. A delay in the closure of these structures would likely affect bulk autophagy of soluble proteins less than the engulfment of organelles and aggregates.

Effective engulfment of such bulky cargos probably requires active mechanical forces to constrict the forming autophagosome. Mechanical constriction could also help to pinch off pieces of cargo, which could explain the ring-like structure observed during mitophagy. ATPase activity of DFCP1 could provide the necessary mechanical forces to effectively close large pre-autophagic structures. In contrast, bulk autophagic structures can be much smaller and consequently require less mechanical forces to close. Thus, DFCP1 might not be necessary to constrict these structures.

In conclusion, we have identified an evolutionarily conserved Dynamin-related ATPase module that controls the early phase of selective autophagy. It is interesting to note that biogenesis of the replication compartments of certain viruses, including SARS-CoV-2, depends on DFCP1[39], and our data thus imply that constriction of ER-derived membranes is a critical step in SARS-CoV-2 replication. ATP binding and hydrolysis of DFCP1 trigger omegasome constriction and thereby the biogenesis of cargo-containing autophagosomes. In future studies it will be interesting to reveal the interplay between DFCP1 and cargoes, and to understand the structural biology of the novel ATPase.

## Methods

### Structure prediction, homology modeling and structural analysis

We performed structure prediction using AlphaFold[11]. Homology modeling was also performed using the Phyre2 server, using DFCP1 amino acids 1–410 as target sequence. The complete output of the Phyre2 modeling and from AlphaFold is provided in the Supplementary Data. For structural comparison with Phyre2, we chose the prediction model with the highest confidence (based on GBP1 (PDB: 1f5n))[40].

All further analysis were performed in UCSF Chimera[41]. The structures of GBP1 and the predicted structures were aligned using the "MatchMaker" function and critical amino acids of the DFCP1 P-Loop domain were identified by their structural proximity to the corresponding amino acids of GBP1. The structural alignment was also used to approximate the localization of the bound nucleotide, shown is the GDP moiety bound by GBP1.

For further comparison to other nucleotide-binding proteins, we generated a structural alignment using the "Match-Align" feature of Chimera, using DFCP1, hGBP1 (PDB:1f5n), hAtlastin1 (PDB: 3q5e), N-RAS Q61H (PDB: 621p) and KRAS (PDB:4obe) as templates.

### Protein purification

Recombinant proteins were expressed as His-tagged MBP fusion proteins in E. coli. Constructs encoding the DFCP1 N-terminus were cloned into pET-His6-MBP-TEV-LIC (Addgene # 29708) and Rosetta2(DE3) bacteria were transformed with the resulting plasmids. For protein expression, bacteria were grown in ZYM505 medium[42]. Expression was induced by 0.25 mM IPTG, and induced cells were grown overnight at 20 °C. Cells were harvested by centrifugation, resuspended in lysis buffer (50 mM Tris pH7.5, 150 mM NaCl, 10 mM MgCl$_2$, 25 mM Imidazole, 1 mM TCEP, cOmplete mini EDTA-free protease inhibitor (Roche)) and lysed by one passage through a homogenizer. Raw lysates were cleared by centrifugation. The His-MBP fusion proteins were purified by affinity chromatography using NiNTA affinity chromatography. Protein-containing fractions were pooled, dialyzed overnight against dialysis buffer (50 mM Tris pH7.5, 150 mM NaCl, 10 mM MgCl$_2$, 10% Glycerol, 1 mM TCEP). Aliquots were snap-frozen in liquid nitrogen and stored at −80 °C.

Full-length DFCP1 was produced as His6-tagged protein in SF9 cells using the Bac-to-Bac baculovirus expression system. Cells harvested by centrifugation and resuspended in high-salt lysis buffer (50 mM Tris pH7.5, 300 mM NaCl, 10 mM MgCl$_2$, 25 mM Imidazole, 1 mM TCEP, cOmplete mini EDTA-free protease inhibitor (Roche)). Cells were lysed using a dounce homogenisator and the raw lysate was cleared by centrifugation. Recombinant DFCP1 was purified by NiNTA affinity chromatography, eluted in high-salt lysis buffer containing 250 mM Imidazole and directly used in ATPase assays.

### Nucleotide-binding assays

Nucleotide-binding assays were carried out by measuring the incorporation of 2,3-0-N-Methyl-anthraniloyl (Mant)-labeled nucleotides (Jena Bioscience) using an Synergy2 plate reader (BioTek Instruments Inc., Winooski, VT, USA) at 25 °C. Reactions containing 1 μM mantNTP, 50 mM Tris pH7.5, 150 mM NaCl, 50 mM MgCl$_2$, 10% Glycerol, 100 μg/ml BSA were allowed to equilibrate for 4 mins. After equilibration, recombinant proteins were added to a final concentration of 2 μM and the fluorescence intensity ($\lambda_{ex} = 360$ nm, $\lambda_{em} = 440$ nm) was measured.

### Measurement of ATPase activity

ATPase activity was measured by quantifying the release of inorganic phosphate using a modified Malachite Green Phosphate assay (Biomol Green, Enzo). All measurements were performed according to the manufacturer's protocol. Briefly, 10 μM purified DFCP1 N-terminus was incubated with 500 μM ATP for 30 min. After 30 min, two volumes of Biomol Green were added and incubated for further 20 min before the OD620 was recorded.

### Membrane-stimulated ATP hydrolysis

Lipid mixtures 70% DOPC, 20% DOPE, 5% DOPS (Avanti Polar lipids) and doped with 5% PtdIns3P (Echelon) or 5% PI (Avanti Polar lipids) were mixed in Chloroform and dried using a stream of nitrogen gas, followed by vacuum. Lipid films were rehydrated with ATPase assay buffer (50 mM Tris-HCL pH7.5, 150 mM NaCl, 10 mM MgCl$_2$, 1 mM TCEP, 10 μM ZnCl$_2$). Liposomes (final lipid concentration: 1 mM) were formed by 5 freeze-thaw cycles. Unilamellar liposomes (100 nm diameter) were generated by extrusion (11×) through polycarbonate filters (100 nm pore size) (Avanti Polar lipids), stored at 4 °C and used within 3 days after formation.

ATPase assays were performed in ATPase buffer (50 mM Tris-HCL pH7.5, 150 mM NaCl, 10 mM MgCl$_2$, 1 mM TCEP, 10 μM ZnCl$_2$) supplemented with 0.5 mM ATP. Liposomes were added to a final concentration of 100 μM lipids. Purified full-length DFCP1 (0.44 μM final concentration) or a corresponding volume of elution buffer was added, and the reaction was incubated for 30 min at 37 °C. After 30 min, the reaction was stopped by addition of two volumes of Biomol Green. The phosphate detection reaction was allowed to develop for further 20 min before the OD620 was recorded.

### Size exclusion chromatography

Size exclusion chromatography was performed using an Äkta Explorer (Cytiva) on a Superdex 200 Increase 10/300 GL column (Cytiva). Recombinant DFCP1 was loaded with nucleotides by incubation with either 400 μM ADP or ATPγS (Jena Bioscience) in exchange buffer (50 mM Tris, 150 mM NaCl, 50 mM MgCl$_2$, 10% Glycerol, 1 mM TCEP) for 20 min at RT, applied to the size exclusion column and then developed by isocratic elution with exchange buffer at 4 °C (flow rate 0.25 ml/min).

## Cell culture

U2OS cells (ATCC: HTB-96), HeLa cells (obtained from Institute Curie, Paris, France), and A431 cells were maintained in DMEM (D0819; Sigma-Aldrich) supplemented with 10% FCS (F7524; Sigma-Aldrich), 5 U/ml penicillin and 50 µg/ml streptomycin at 37 °C with 5% $CO_2$. hTERT-RPE-1 cells (ATCC: CRL-4000) were grown in DMEM/F12 medium (31331-028; Gibco-BRL) with 10% FCS, 5 U/ml penicillin and 50 µg/ml streptomycin at 37 °C with 5% $CO_2$. Parental cell lines and its derivatives were regularly tested for mycoplasma. For starvation experiments, cells were washed twice with EBSS (24010-043; Gibco-BRL) and incubated in EBSS for the indicated time. Bafilomycin A1 (Enzo life sciences, BML-CM-110-0100) was added at 100 nM, Puromycin at 2.5–10 µg/ml (Sigma, P8833), DFP (#379409, Sigma-Aldrich) at 0.5–1 mM, and Torin1 (#4247, Bio-Techne R&D systems Europe) at 50 nM.

## Generation of cell lines

Cell lines stably expressing constructs were generated lentiviral transduction using a third-generation lentiviral vector system as previously described[43]. Subsequently, cells were selected for integration of the expression cassette with the following antibiotics: Puromycin (1 µg/ml), Blasticidin (10 µg/ml), or Geneticin (500 µg/ml). All lentiviral constructs were expressed from a phospho-glycerate kinase (PGK) promoter.

## Generation of DFCP1 KO cell line with CRISPR/Cas9

Deletion of DFCP1 in parental U2OS and A431 cells was generated using the CRISPR/Cas9 system. Two guide RNAs were used due to the presence of internal start codons in Exons 5–7 after the primary start codon in Exon2. The guide RNAs were designed in Benchling (http://www.benchling.com). gRNA1 '5- TGTCCCTTACTGTGACCTCT-3' binds after the primary start codon in Exon2, gRNA2 '5- TGGCGTGGTCTATCG-TAGT-3' binds in Exon8. The guide RNAs were cloned into a pX459 plasmid encoding Cas9-2A-Puro, and the final construct was transfected into the cells with FugeneHD (U2OS) or Lipofectamine LTX with Plus reagent (A431). After 48 h post-transfection cells were selected with Puromycin. Single colonies were picked and characterized. Clones lacking DFCP1 were identified by western blotting, and the genetic changes were characterized by PCR and sanger sequencing. Genetic changes sequences were analyzed by using the Synthego ICE software package (https://ice.synthego.com/#/).

## Generation of DFCP1 KI cell line with CRISPR/Cas9

U2OS cells expressing endogenously tagged DFCP1 were generated by using CRISPR/Cas9 in conjunction with an AAV-based homology donor. Cells were first transduced with the AAV harboring the homology donor, and electroporated 16 h later with CRISPR/Cas9 RNP particles using a Nepa21 electroporator.

Recombinant Cas9, crRNA and tracrRNA were purchased from IDT and RNP complexes were formed according to the manufacturer's manual. The gRNA was located at the start codon of DFCP1 (gRNA sequence: CTGGGCACTCATACTCACGC).

The tagging strategy was based on a two-step tagging technique[44] using a tagging cassette based on the plasmid FLAG-LoxP-PURO-LoxP-HaloTERT WT HR Donor (Addgene # 86843), with the Halo-Tag exchanged by EGFP. This technique first integrates a tagging construct including a resistance marker, which is excised by Cre-mediated recombination in a second step. To generate the homology donor, a construct containing ~1KB of homology left and right of the start codon and the tagging was assembled in an AAV vector (pAAV-2Aneo v2, Addgene plasmid 800333) and packaged using pHelper and pRC2-miR321 vectors (Clontech). AAV particles were isolated using AAVPro extraction solution (Clontech). After transduction with both CRISPR/Cas9 gesicles and AAV homology donor, cells with integrated homology donor were selected using puromycin selection (1 µg/ml). Single clones were picked and characterized by PCR and western blotting and

the integration site verified by sequencing. Successful homozygous clones were then treated with Cre recombinase to excise the resistance marker, resulting in a cell line expressing FLAG-EGFP-DFCP1 under control of the endogenous promoter.

## siRNA-mediated knockdown

For siRNA transfections, 20 nM of siRNA was transfected using Lipofectamine RNAiMax reagent. Transfected cells were analyzed 48–72 h after transfection; knockdown efficiency was verified by western blotting. For knockdown of DFCP1 or ATG13, the following Silencer Select siRNAs from Ambion (Thermo Fisher Scientific, Waltham, MA, USA) were used: Silencer® Select Negative Control No.1 (4390843), siDFCP1-1 (ID s28712), siDFCP1-2 (ID s28714), siDFCP1-3 (ID 135291), or siATG13 (ID s18879).

## Immunoblotting

U2OS and RPE cells were washed with ice-cold PBS and lysed in 2.8× Laemmli Sample buffer (Bio-Rad) containing 200 mM DTT. Cell lysates were subjected to SDS-PAGE on a 4–20% (567-1094; Bio-Rad) gradient gel and blotted onto PVDF membranes (Bio-Rad). Membranes were probed with primary antibodies overnight at +4°C. Membranes incubated with fluorescently labeled secondary antibodies were developed by Odyssey infrared scanner (LI-COR). Membranes detected with HRP-labeled secondary antibodies were developed using Clarity Western ECL substrate solutions (Bio-Rad) with a ChemiDoc XRS+ imaging system (Bio-Rad).

A431 cells were washed with ice-cold PBS and lysed into RIPA buffer (1% IGEPAL CA-630, 0.1% SDS in TBS) containing protease inhibitors. Protein concentrations were measured using DC protein assay and 5× Laemmli Sample Buffer with mercaptoethanol was added. Samples were heated to 95 °C for 5 min and then blotted as above. Equal amounts of protein were loaded onto 10% Mini-PROTEAN TGX Stain-Free gels and transferred onto PVDF membrane. Membranes were blocked with 5% milk in TBS containing 0.1% Tween-20 for 1 h at RT and probed with primary antibodies at +4°C overnight. After washing with TBS containing 0.1% Tween-20, membranes were incubated with HRP-labeled secondary antibody for 1–2 h at RT. Membranes were washed, incubated with ECL Clarity Max substrate, and imaged with a ChemiDoc Imaging System (Bio-Rad). Stain-Free total protein signals are shown as loading controls in Supplementary Fig. S9h. Unprocessed and full scan blots are shown in the source data file.

The following primary antibodies have been used for immunoblotting: anti-DFCP1 (rabbit, Cell Signaling Technology, mAb #85156, Clone E9R6P, 1:500); anti-GFP (mouse, Roche, 11814460001, Clone 7.1 and 13.1, 1:500); anti-beta-actin (mouse, Sigma-Aldrich, A5316, Clone AC-74, 1:20,000); anti-LC3B (rabbit, Cell Signaling Technology, 2775s, 1:1000); anti-p62 (rabbit, MBL, PM045, 1:10,000); anti-Atg13 (rabbit, Cell Signaling Technology, #13468, Clone E1Y9V, 1:1000).

The following secondary antibodies have been used for immunoblotting: goat anti- rabbit IgG HRP (Bio-Rad, 1706515, 1:2000); goat anti-rabbit HRP (Jackson, 111-035-144, 1:5000); goat anti-mouse HRP (Jackson, 115-035-003, 1:5000); donkey anti-rabbit IRDye 680RD (LI-COR Biosciences, 926-68073, 1:5000); donkey anti-mouse IRDye 680RD (LI-COR Biosciences, 926-68072, 1:5000); donkey anti-rabbit IRDye 800CW (LI-COR Biosciences, 926-32213, 1:5000); donkey anti-mouse IRDye 800CW (LI-COR Biosciences, 926-32212, 1:5000).

## Measuring autophagy by ratiometric flow cytometry

The monomeric Keima (mKeima) protein is a useful tag for measuring autophagic degradation of various autophagic cargoes, such as mitochondria[24]. Since mKeima is resistant to lysosomal proteases and has a bimodal, pH-responsive excitation spectrum, the assay provides a cumulative readout of autophagic activity as mKeima stably accumulates and undergoes a change in excitation maximum upon

trafficking to the acidic environment of lysosomes[25,45]. A probe for mitophagy was obtained by fusing the mitochondrial-targeting presequence of COX VIII to mKeima, producing mito-mKeima. To compare mitophagy with bulk autophagy of cytosolic proteins, we used either free mKeima itself or the cytosolic protein lactate dehydrogenase (LDHB) fused to mKeima as cytosolic probes[24]. RPE-1 cells expressing LDHB-mKeima or mito-mKeima[31] and U2OS cells expressing free mKeima or mito-mKeima in an inducible manner were generated by lentiviral transduction. The cells were treated with doxycycline (200 ng/ml) to induce expression of the mKeima probes during the 2 days of siRNA-mediated depletion of DFCP1 or ATG13. Before further treatments, the cells were washed twice to remove doxycycline. The cells were subsequently treated with EBSS, Torin1 (50 nM) or DFP (0.5 mM) for 16 h, detached by trypsin/EDTA and resuspended in complete cell culture medium for analysis on a BD LSR II Flow Cytometer (BD Biosciences, San Jose, CA, USA) connected to the BD FACSDiva™ software (BD Biosciences, San Jose, CA, USA). Autophagy was determined as the ratio of the median fluorescent intensity of the mKeima signal excited by 561 nm (45 mW) laser divided by mKeima excited by 407 nm (100 mW) laser, with a 610/20 bandpass filter and a 600 nm long pass dichroic filter in both cases. The gating strategy has been described previously[46,31]. By using the FlowJo™ software (BD Biosciences, San Jose, CA, USA), the derived ratio of 561/407 nm signal intensity per cell was obtained, and the median value of these cellular ratios were compared between treatments.

### Puromycin treatment to induce aggrephagy
Puromycin (Sigma, P8833) was added at a concentration of 2.5 or 10 μg/ml to complete medium, cells were incubated for the indicated time, fixed, and stained as described below.

### Immunofluorescence microscopy and image analysis
Cells were seeded on glass coverslips, fixed with 4% formaldehyde (FA; 18814; Polysciences) for 12 min at room temperature, and permeabilized with 0.05% saponin (S7900; Sigma-Aldrich) in PBS. Fixed cells were then stained with primary antibodies at room temperature for 1 h in PBS/saponin, washed in PBS/saponin, stained with fluorescently labeled secondary antibody for 1 h, washed in PBS, and mounted with Mowiol containing 2 μg/ml Hoechst 33342 (H3570; Thermo Fisher Scientific).

The following primary antibodies have been used: anti-mNeonGreen (mouse, Chromotek, 32f6-100, Clone 32F6, 1:500); anti-LC3B (rabbit, MBL, PM036, 1:500); anti-LC3B (rabbit, Cell Signaling Technology, 2775 s, 1:100); anti-WIPI2 (mouse, abcam, ab105459, Clone 2A2, 1:200); anti-p62 (rabbit, MBL, PM045, 1:500); anti-p62 (guinea pig, Progen, GP62-C, 1:250); anti-Ubiquitin (mouse, EMD Millipore Corp., 04-263, Clone FK2, 1:400); anti-Ubiquitin-K63 (rabbit, Millipore, 05-1308, Clone Apu3, 1:200), anti-Tom20 (mouse, BD, 612278, Clone 612278, 1:100).

For secondary antibodies we used donkey Alexa488-anti-mouse (Jackson, 715-545-151, 1:500); donkey Alexa488-anti-rabbit (Jackson, 711-545-152, 1:500); donkey Alexa568-anti-mouse (Molecular Probes, A10037, 1:500); donkey Alexa568-anti-rabbit (Molecular Probes, A10042, 1:500); donkey Alexa647-anti-mouse (Jackson, 715-605-150, 1:500); donkey Alexa647-anti-rabbit (Jackson, 711-605-152, 1:500); donkey Alexa647-anti-guinea pig (Jackson, 706-605-148, 1:500).

Confocal micrographs were obtained using an LSM710 confocal microscope (Carl Zeiss) equipped with an Ar-laser multiline (458/488/514 nm), a DPSS-561 10 (561 nm), a continuous-wave laser diode 405–30 CW (405 nm), and an HeNe laser (633 nm). The objective used was a Plan-Apochromat 63×/1.40 oil differential interference contrast (DIC) III (Carl Zeiss). For visualization, images were analyzed and adjusted (brightness/contrast) in ImageJ/Fiji[47] or Zen Blue.

For quantification, all images within one dataset were taken at fixed intensities below saturation, very high expressing cells have been omitted, and identical settings were applied for all treatments within

one experiment. In general, 25 images were taken randomly from each condition.

The NIS-elements software was used for background correction ("rolling ball") and automated image analyses. Identical analysis settings were applied for all treatments within one experiment. Fluorescent LC3B, p62 or WIPI2 dots were segmented by the software, and the number and sum fluorescence intensity of the dots were measured, as well as the co-occurrence of WIPI2 and DFCP1 dots. The total number of cells was quantified by automated detection of Hoechst nuclear stain by the software. The number of mNG-DFCP1 omegasome rings was quantified manually, scoring all objects with a visible lumen. Micronuclei were quantified manually from micrographs.

For widefield imaging of micronuclei in A431 cells, cells were grown on Ibidi m-slide 8 well ibiTreat chambers. Cells were then washed with PBS and fixed with 4% PFA in 250 mM HEPES (pH 7.4), 100 mM $CaCl_2$, and 100 mM $MgCl_2$ for 20 min, followed by quenching in 50 mM $NH_4Cl$ for 10 min and 3 washes with PBS. Staining with DAPI (Sigma, D9542, 10 μg/ml final concentration in PBS) was done for 10 min at RT, followed by 2 washes with PBS. Z-stacks covering the whole cell at 0.3 μm spacing were acquired with Nikon Eclipse Ti-E microscope equipped with a 60× Plan-Apochromat λ oil objective, NA 1.4. The number of nuclei and micronuclei per maximum intensity projected z-stack images were manually counted in ImageJ/Fiji.

### Live Imaging of omegasome formation
Live-cell imaging was performed on a Deltavision OMX V4 microscope equipped with three PCO.edge sCMOS cameras, a solid-state light source and a laser-based autofocus. Environmental control was provided by a heated stage and an objective heater (20–20 Technologies). Images were deconvolved using softWoRx software and processed in ImageJ/FIJI. Cells were imaged in EBSS, and images were taken every 2 s over a period of 10–15 min. For imaging of SNAP-LC3B, cells were stained with SiR647-SNAP (New England Biolabs) according to the manufacturer's protocol.

### Tracking of omegasomes
Omegasomes were initially tracked manually in Fiji. For each timepoint of each track, a ROI of 50 × 50 pixels was extracted, the DFCP1 signal was thresholded using the Otsu algorithm and particles larger than 3 pixels were further analyzed using the "Analyze particles" tool in Fiji. To exclude random particles moving into the field of view, only structures within 10 pixels of the center of the ROI were considered. We measured the mean intensity, the area and the Feret's diameter of each particle over time. Data was then postprocessed in python and the Ferets diameter was plotted using Seaborn. For the interpretation of these plots, we defined the maximum diameter of the omegasome as the starting point of constriction, and the disappearance of the omegasome was defined as fully constricted omegasome.

Scripts for tracking, measuring and postprocessing are supplied at the following github address: https://github.com/koschink/Naehse_et_al.

### Correlation of omegasome size and closure
Data analysis was performed using the Python "pandas" package, and the maximum omegasome diameter and the length of each track was extracted. The "SciPy" statistics package was used to calculate the Spearman correlation, and data was plotted using Seaborn.

### Structured illumination microscopy (SIM)
Cells were prepared according to the immunofluorescence protocol detailed above. Three-dimensional SIM imaging was performed on Deltavision OMX V4 microscope with an Olympus ×60 NA 1.42 objective and three PCO.edge sCMOS cameras and 488 nm, 568 nm and 647 nm laser lines. Cells were illuminated with a grid pattern and for each image plane, 15 raw images (5 phases and 3 orientations) were

acquired. Super-resolution images were reconstructed from the raw image files and aligned using softWoRx software (Applied Precision, GE Healthcare). Images were processed in ImageJ/Fiji. 3D reconstructions were performed using Imaris.

## Statistics

The number of individual experiments and the number of cells or images analyzed are indicated in the figure legends. The number of experiments was adapted to the expected effect size and the anticipated consistency between experiments. We tested our datasets for normal distribution by Kolmogorov–Smirnov, D'Agostino and Pearson, and Shapiro–Wilk normality tests, using GraphPad Prism Version 8. For parametric data, an unpaired two-sided $t$-test was used to test two samples with equal variance, and a one-sample $t$-test was used in the cases where the value of the control sample was set to 1. For more than two samples, we used ordinary one-way ANOVA with a suitable post hoc test. For nonparametric samples, Mann–Whitney test was used to test two samples and Kruskal–Wallis with Dunn's post hoc test for more than two samples. All error bars denote mean values ± SD or SEM, as indicated in every figure legend (*$P < 0.05$; **$P < 0.01$; ***$P < 0.001$). No samples were excluded from the analysis.

## Reporting summary

Further information on research design is available in the Nature Portfolio Reporting Summary linked to this article.

## Data availability

All data shown and used to generate plots, as well as detailed statistical information, accompanies this manuscript in the source data file. Uncropped and unprocessed western blots and gels are shown in source data files. Underlying image data are available from the corresponding authors (V.N., K.O.S, H.S.) upon request. Publicly available entries used in this study are PDB:1f5n, PDB: 3q5e, PDB: 621p, PDB:4obe. Source data are provided with this paper.

## Code availability

Scripts for tracking, measuring and postprocessing are supplied at the following github address: https://github.com/koschink/Naehse_et_al.

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

## Acknowledgements

We thank the Flow Cytometry Core Facility and the Advanced Light Microscopy Core Facility of Oslo University Hospital, and the Helsinki Institute of Life Science (HiLIFE) Light Microscopy Unit for technical assistance and access to instruments. Andreas Brech and the Advanced Electron Microscopy at Oslo University Hospital are acknowledged for expert help with electron microscopy sample processing. We thank Ulrikke Dahl Brinch for general technical help and the entire Stenmark group for discussion. V.N. was supported by a Mobility grant from the Research Council of Norway (grant number 301369). C.R. was supported by the Norwegian Cancer Society (project number 198140). M.L.T. was supported by the Research Council of Norway (grant number 274574). V.T.S. was supported by Biomedicum Helsinki Foundation. M.N. was supported by a Grant from the Research Council of Norway (grant number 249884) to T.J. E.M.W was supported by the Radium Hospital Foundation (InvaCell grant / Trond Paulsen). K.O.S was supported by a Career grant from the South-Eastern Norway Regional Health Authority (grant number 2020038) and a Research Grant from the Research Council of Norway (grant number 315103). H.S. was supported by Project Grants from the South-Eastern Norway Regional Health Authority (project number 2018081) and the Norwegian Cancer Society (project number 182698), and by an Advanced Grant from the Europan Research Council (project number 788954). This work was partly supported by the Research Council of Norway through its Centres of Excellence funding scheme, project number 262652.

## Author contributions

V.N and K.O.S. conceived and designed the study with contributions from C.R. and H.S. V.N. generated cell lines, performed light microscopy and biochemical assays, analyzed data, prepared figures, and wrote the original draft. K.O.S. performed structural analyses, biochemical assays, prepared figures and edited the draft. C.R. performed HiC imaging, analyzed data, prepared figures, and edited the draft. K.W.T. generated cell lines, performed HiC imaging, analyzed data, and prepared figures. M.N. and T.J. performed and analyzed GST binding assays, prepared figures and edited the draft. M.L.T. performed the mKeima assays, analyzed data and prepared figures. S.M. generated KO and KI lines, performed SIM, analyzed data, and prepared figures. E.M.W. performed immunoprecipitation experiments, analyzed data, and prepared figures. V.T.S. and E.I. performed and analyzed experiments in A431 cells and edited the manuscript. H.S. supervised the study, contributed to funding acquisition and research infrastructure, and edited the manuscript.

## Competing interests

The authors declare no competing interests.
