## [Peer Review File · Nature Communications]

ATPase activity of DFCP1 controls selective autophagyREVIEWER COMMENTS

Reviewer #1 (Remarks to the Author):

In this manuscript, the authors have addressed the role of DFCP1 in autophagy and in particular the mechanisms for how it impacts omegasome maturation. The presented data show that DFCP1 is an ancient protein that binds and hydrolyses ATP and that ATP-binding facilitate dimerization of DFCP1. The study identifies and characterizes specific mutants defective in ATP-binding or hydrolysis based on homology to other GTP/ATPases. Using live cell SIM analysis, the authors demonstrate the temporal activity of DFCP1 at omegasomes in relation to other autophagy components. Using the DFCP1-mutants ATP-hydrolysis is shown to be important for the omegasome constriction stage. Furthermore, the results show that depletion or mutation of DFCP1 affects selective, but not bulk, autophagy. The work is original, and although DFCP1 is a known component of the omegasome, this work significantly contributes to our understanding of the mechanism of DFCP1, but also the process of selective autophagy. Characterization of DFCP1 and the impact of specific DFCP1-mutants has the potential to majorly impact future work on autophagy since this provides tools to delay the process of omegasome constriction. This also advances our understanding of the large family of NTPases involved in membrane remodelling processes. The manuscript is well written, figures are nicely illustrated and the presentation of the work is very clear and coherent. The supplementary data is impressive and clearly supports the main figures. The authors have used novel tools and state-of-the-art imaging techniques. The experimental set up is well thought through and controls are sound. The authors use a very nice KO-based complementation system. The presented results support most of the conclusions in the manuscript, although I have some specific comments to some of the conclusions (see below) and claims. Overall, the work and manuscript provide a significant advance to the field.

Specific points raised:

-Characterisation of DFCP1 as an ATPase is very important. Binding of ATP is described as “rapid”, as compared to what? Analysis of any control ATPase within the same large family would strengthen this statement. Binding of ADP appears to be similar to that of ATP. Can something be stated regarding affinities, since this might be important? This would also be good in order to compare to other ATPases.

-Comparing the ATPase activity assay in Fig 1e and Fig1g, there is a larger difference between DFCP1 and MBP in Fig1e, but in Fig1g the difference is much smaller. Does this indicate that the ATPase activity is quite low? Again, incorporation of a control ATPase control would clarify this.

-The manuscript proposes that DFCP1 works in a similar fashion as large NTPases such as Dynamin, Atlastin, EHD2 etc. A key feature of some of these is that the NTPase activity is stimulated following

membrane binding. Due to the proposed mechanism of DFCP1 in membrane constriction, I think it would be relevant to analyse if the GTPase activity is stimulated by membranes.

-The authors show that dimerization is dependent on ATP, but independent of membranes. In the model proposed in supplementary fig9, DFCP1 is illustrated as oligomers and the proposed mechanism for membrane constriction involves oligomers. To support this, it could be relevant to address oligomerisation for example via an in vivo FRET assay in the presence of membranes, or by FLIM analyses of fluorescently tagged DFCP1 in cells. This would strengthen the proposed model.

-Based on the described work in Fig 2, omegasome constriction is delayed in cells expressing ATPase deficient DFCP1. Yet, it still constricts as I understand. The reasons for this would be interesting to elaborate on in the discussion. This is also in agreement with that the difference in p62 accumulation in fig3 is rather small following mutant expression or DFCP1 depletion

-Would the DFCP1 activity to constrict the omegasome be similar in aggregophagy, mitophagy and microneurophagy?

-In Fig4i (also in supp fig 8), the authors analyse binding of DFCP1 to Ubiqu-proteins. These data are not so convincing since it is based on immunoprecipitation where the difference, as compared to control, is not major. If there is a direct interaction between DFCP1 and ubiqu this should be possible to address in vitro since the authors have purified proteins. On the other hand, I don't understand why ubiqu-binding or not is important for the presented story. It would make more sense to investigate this firmly in a separate story so maybe this could be removed.

Reviewer #2 (Remarks to the Author):

This manuscript proposes that the PI3P binding protein DFCP1 consumes ATP to shape autophagosome formation, akin to how Dynamins perform membrane scission through GTP hydrolysis. DFCP1 is a highly-used imaging marker for following autophagosome formation, but very little is known about its function. Elucidation of how DFCP1 works at the molecular level will be invaluable to the autophagy field. The model put forward by the authors is intriguing, but the experiments need to be better designed to support the claims made in the manuscript.

Major points:

1. In Fig. 1, the authors measured DFCP1 ATP-hydrolysis without PI3P-rich membranes. This design begs the question- is the activity measured related to DFCP1's function on PI3P-rich subdomains of the ER? Based on the model presented in Extended Fig. 9, it appears that DFCP1 should only drive ATP hydrolysis during membrane constriction?
2. In Fig. 2b, the authors imaged mNG-DFCP1 and mCh-p62 and suggested that DFCP1 couples ATP hydrolysis to membrane constriction. While the data can be used to phenomenologically show that there are three "phases" (initiation, maturation, and constriction) in omegasome (DFCP1-labeled) mediated autophagosome formation, it is difficult to see how one can clearly define the boundaries between the three phases in the images (e.g., why is the boundary between the initiation and maturation phases at $t=80s$ in the example shown?). Furthermore, it seems that the images were manually inspected and assigned. This method of quantification can make the results presented in Fig. 2c unreliable.
3. Overexpression of DFCP1 is known to block autophagosome formation (PMID: 18725538). This observation means that alteration of DFCP1 levels will lead to changes in the dynamics of autophagosome formation. In this paper, the authors used stable cell lines expressing lower levels of mNG-DFCP1 (WT, K193A, and T189A) for imaging. The authors should carefully quantify and list the expression levels of DFCP1 in these lines (shown in Extended Fig. 3c), as they are used to report minute changes in omegasome formation time in Fig. 2c (from $t\sim 350s$ for WT to $t=400-500s$ for the mutants).
4. From Figs. 1-3 the authors utilized cells under starvation (to induce autophagy) to assay for the functions of DFCP1. However, it has been shown that DFCP1 depletion does not affect the autophagy flux during starvation-induced autophagy (PMID: 18725538). To get out of this apparent paradox, the authors between LINE 186-201 appeared to invoke the argument that only a small fraction of autophagosomes formed during starvation come out of DFCP1-labeled omegasomes. The authors need to directly demonstrate this claim as it does not match the results from the existing literature.
5. The authors suggested that DFCP1 helps the formation of autophagosomes around ubiquitinated substrates. Therefore, it is confusing why they looked at the role of DFCP1 in DFP-induced mitophagy. It is unclear whether ubiquitination is needed during DFP-induced mitophagy. It makes more sense for the authors to demonstrate how DFCP1 affects ubiquitin-dependent mitophagy.
6. The authors suggested that DFCP1 can directly bind to poly-ubiquitinated proteins, allowing it to mediate autophagosome formation around ubiquitinated substrates specifically. This claim is not sufficiently developed. One may also counter-argue that direct binding between DFCP1 and the substrates can impede autophagosome formation: LC3-labeled autophagosomes will need to compete with DFCP1 for substrate association.

Reviewer #3 (Remarks to the Author):

In this manuscript, the authors demonstrate an important role of DFCP1 in autophagy as an ATPase, which relies on its ability to bind and hydrolyse ATP. Specifically, DFCP1 regulates the formation of autophagosomes. The DFCP1 mutants with reduced ATP binding or hydrolysis delay the release of nascent autophagosomes from omegasomes, resulting in p62 accumulation. Multiple advanced techniques such as super-resolution imaging have been employed to comprehensively study the new role of DFCP1. The paper is well written. Several major issues have to be addressed.

1. The authors define several stages of omegasome dynamics mainly based on the SIM images. The choice was made by the eyes, which would be very arbitrary. There should be a more quantitative or definitive way to segregate these stages. At least, the authors should show more representative SIM images with clear indicated features for staging.
2. For to the “initiation” stage, “characterized by a DFCP1 spot which is formed de novo, and to which p62 and LC3B are recruited a few seconds later”, the authors missed one critical parameter about the initiation, which is the successful rate of recruiting p62 and LC3B. Clearly, not every DFCP1 can lead to the formation of omegasome. Therefore, this successful rate would be very important to evaluate the function of DFCP1, especially, when the authors couldn't see a change on initiation and maturation for DFCP1 K193A or T189A.
3. The DFCP1 K193A and T189A mutants only delay omegasome formation. Basically, functional omegasomes can still form for autophagy. Therefore, it's not conclusive to say that delayed omegasomes without defects are directly related to p62 accumulation unless the number of omegasome formation for these two mutants is similar to what wild-type does.
4. In order to conclude that the interaction between DFCP1 and ubiquitin would be influenced by the mutants K193A or T189A. More quantification should be conducted. For instance, a co-IP experiment similar to Fig. 4i should be performed.
5. In the discussion section, the authors speculate that DFCP1 works on the membrane to regulate omegasome, similar what dynamin does. Dynamin is known for membrane tubulation [<https://www.nature.com/articles/nrm1313>]. Therefore, the authors should check the capability of DFCP1 and its mutants on membrane tubulation. This would significantly enhance the impact of this paper.

REPLIES TO REVIEWER COMMENTS

Reviewer comments are in black font, and our replies in blue.

We thank all three reviewers for their very insightful and helpful comments, which have certainly contributed to a profound improvement of our manuscript. Among other revisions, we have now purified full-length DFCP1 and show that its ATPase activity is stimulated by membrane binding, and we have determined the K_D values for ATP and ADP. We have prepared a script for unbiased quantifications of omegasome size and used this to confirm that DFCP1 mutants defective in ATP binding and hydrolysis cause stalled constriction of omegasomes. As suggested by the reviewers, we have removed data on ubiquitin interactions. Instead, we now present data indicating that DFCP1 is selectively required for constriction of large omegasomes, which provides a plausible explanation why it is required for selective but not for bulk autophagy. We have thus amended our model for DFCP1 and omegasome functions in selective autophagy.

Reviewer #1:

In this manuscript, the authors have addressed the role of DFCP1 in autophagy and in particular the mechanisms for how it impacts omegasome maturation. The presented data show that DFCP1 is an ancient protein that binds and hydrolyses ATP and that ATP-binding facilitates dimerization of DFCP1. The study identifies and characterizes specific mutants defective in ATP-binding or hydrolysis based on homology to other GTP/ATPases. Using live cell SIM analysis, the authors demonstrate the temporal activity of DFCP1 at omegasomes in relation to other autophagy components. Using the DFCP1-mutants ATP-hydrolysis is shown to be important for the omegasome constriction stage. Furthermore, the results show that depletion or mutation of DFCP1 affects selective, but not bulk, autophagy. The work is original, and although DFCP1 is a known component of the omegasome, this work significantly contributes to our understanding of the mechanism of DFCP1, but also the process of selective autophagy. Characterization of DFCP1 and the impact of specific DFCP1-mutants has the potential to majorly impact future work on autophagy since this provides tools to delay the process of omegasome constriction. This also advances our understanding of the large family of NTPases involved in membrane remodelling processes. The manuscript is well written, figures are nicely illustrated and the presentation of the work is very clear and coherent. The supplementary data is impressive and clearly supports the main figures. The authors have used novel tools and state-of-the-art imaging techniques. The experimental set up is well thought through and controls are sound. The authors use a very nice KO-based complementation system. The presented results support most of the conclusions in the manuscript, although I have some specific comments to some of the conclusions (see below) and claims. Overall, the work and manuscript provide a significant advance to the field.

Specific points raised:

- 1) Characterisation of DFCP1 as an ATPase is very important. Binding of ATP is described as “rapid”, as compared to what? Analysis of any control ATPase within the same large family would

strengthen this statement. Binding of ADP appears to be similar to that of ATP. Can something be stated regarding affinities, since this might be important? This would also be good in order to compare to other ATPases.

We agree with the reviewer and have now strengthened the characterization of the DFCP1-ATPase activity by new experiments. Importantly, we have determined the equilibrium dissociation constant (K_D) of DFCP1 for ATP-binding using both mantATP and mantADP and increasing doses of protein (New Fig1e). The calculated K_D of 10 μ M (ATP) and 5 μ M (ADP) is in the same range as the well characterized large ATPase involved in membrane remodeling, EHD2, with a K_D of 13 μ M for ATPyS and 50 μ M for ADP (Daumke, O et al, Nature 2007).

Our conclusion that the binding of ATP by DFCP1 is 'rapid' is based on the comparison to the GTP-binding properties of Cdc42, where the maximum binding of GTP has not been reached for the time of measurement (60 min) (Fig. 1d). The maximum binding of ATP by DFCP1 is reached within 5-10 min after adding the nucleotide (Fig. 1c). We apologize that this was not stated clearly in the previous version of the manuscript. This is now emphasized in the text, page 3.

- 2) Comparing the ATPase activity assay in Fig 1e and Fig1g, there is a larger difference between DFCP1 and MBP in Fig1e, but in Fig1g the difference is much smaller. Does this indicate that the ATPase activity is quite low? Again, incorporation of a control ATPase control would clarify this.

We found that the background of measurements varied strongly due to aging of the phosphate detection solution used in the assays (Biomol Green). In order to minimize this variance, we have repeated the ATPase activity assays using a fresh malachite-green based (Biomol Green) solution (New Fig1k). Using fresh Biomol solution, the MBP control has been consistently low. The activity of DFCP1 as compared to the control was similarly high in the new vs. the original experiments.

- 3) The manuscript proposes that DFCP1 works in a similar fashion as large NTPases such Dynamins, Atlastins, EHD2 etc. A key feature of some of these are that the NTPase activity is stimulated following membrane binding. Due to the proposed mechanism of DFCP1 in membrane constriction, I think it would be relevant to analyse if the GTPase activity is stimulated by membranes.

We thank the reviewer for this excellent suggestion. To answer this important question, we purified full-length DFCP1 from insect cells and incubated with ATP in the absence or presence of liposomes, which either contain PI3P or PI. The released inorganic phosphate was measured with a malachite green based reagent. Adding liposomes led to a highly significant increase in organic phosphate (New Fig.2c). Thus, the presence of membranes increases the ATPase activity of DFCP1.

- 4) The authors show that dimerization is dependent on ATP, but independent of membranes. In the model proposed in supplementary fig9, DFCP1 is illustrated as oligomeres and the proposed mechanism for membrane constriction involves oligomers. To support this, it could be relevant to address oligomerisation for example via an in vivo FRET assay in the presence of membranes, or by FLIM analyses of fluorescently tagged DFCP1 in cells. This would strengthen the proposed model.

The main support for a multimerization of DFCP1 is derived from our gel filtration experiments. However, these measurements only included the isolated DFCP1 N-terminus including the NTPase domain. This

recombinant protein lacks the membrane association domain. We found that dimerization is dependent on the NTPase domain and – based on structural modelling, likely occurs in a “head-to-head” fashion, similar to Dynamin and EHD. This dimerization mode indeed only accounts for a potential dimer, and our structural modelling did not reveal any other clear alternative dimerization domains like the “stalk” region of Dynamin.

While the suggested FRET and FLIM measurements would be interesting, the ability of DFCP1 to form dimers would always result in FRET *in vitro* (unless one assumes that dimerization is irreversible), whereas cellular measurements would suffer from bystander FRET (which also affects FLIM measurements). We attempted to show multimerization on liposomes by negative staining EM, but so far we have not identified robust conditions for this. We have therefore weakened the claims in the model and clearly indicate speculative parts.

- 5) Based on the described work in Fig 2, omegasome constriction is delayed in cells expressing ATPase deficient DFCP1. Yet, it still constricts as I understand. The reasons for this would be interesting to elaborate on in the discussion. This is also in agreement with that the difference in p62 accumulation in fig3 is rather small following mutant expression or DFCP1 depletion

The reviewer is correct, omegasomes are delayed in their late constriction, but they do constrict. The process until a visible omegasome ring appears is similar in WT and mutants, meaning omegasome initiation is not affected. In addition to our previous manual quantification of omegasome formation and constriction (Fig.3g, h, ExtDataFig.4e-g), we have analyzed an entire new data set using a script for automated quantification (New Fig.3i). Using this automated image analysis, we could confirm our previous data: the time from ring-formation to the complete disappearance of the DFCP1 signal is delayed in the mutants. As the mutants are rather delayed, but not fully stalled, p62 accumulates accordingly and its presence at the omegasome is prolonged together with DFCP1, leading to a total increase of DFCP1 and p62 puncta.

Since the mutants do eventually constrict, it is likely that other factors than DFCP1 are involved in constriction. It is conceivable that the initial constriction after ring formation is governed by another factor, and that DFCP1 performs the remaining constriction when the diameter is smaller/the curvature higher. To test this hypothesis and identify such factors, however, goes beyond the scope of this study. As the reviewer suggests, we have elaborated on this in the Discussion page 11.

- 6) Would the DFCP1 activity to constrict the omegasome be similar in aggrephagy, mitophagy and micronucleophagy?

Morphology-wise, omegasomes at micronuclei, mitochondria or ER (data not shown) look similar, as they go through the same stages as omegasomes induced by EBSS, and they have a fairly similar size range. Therefore, we do not expect that constriction is different in the different types of selective autophagy.

Consistent with this, inducible formation of protein aggregates (Janssen, AFJ et al, JCS 2021, <https://doi.org/10.1242/jcs.258824>) led to DFCP1 recruitment, and the formation of a ring structure that stayed at the cargo with a similar time frame as we observed , 6-10min.

In another study (Zachari M et al, Dev Cell 2019 <https://doi.org/10.1016/j.devcel.2019.06.016>), where mitophagy was induced by Ivermectin, a DFCP1 ring-structure appeared at mitochondria, and the entire omegasome persisted approximately 8-10min.

Thus, the omegasome morphology seems to be independent of the cargo, and constriction happens likely as a similar process. We thank the reviewer for raising this point, which is now discussed in the manuscript on page 11/12.

- 7) In Fig4i (also in supp fig 8), the authors analyse binding of DFCP1 to Ubiquitin-proteins. These data are not so convincing since it is based on immunoprecipitation where the difference, as compared to control, is not major. If there is a direct interaction between DFCP1 and ubiquitin this should be possible to address in vitro since the authors have purified proteins. On the other hand, I don't understand why ubiquitin-binding or not is important for the presented story. It would make more sense to investigate this firmly in a separate story so maybe this could be removed.

We agree with the reviewer, that in vitro studies are needed to address the binding between DFCP1 and ubiquitin-chains. We have not been able to detect any direct interaction between DFCP1 and ubiquitin in such experiments, so we have decided to remove the ubiquitin binding data and discussion in the revised manuscript. Instead, we speculate that the relative importance of DFCP1 for selective vs. bulk autophagy could be related to the larger size of the former autophagosomes.

Reviewer #2:

This manuscript proposes that the PI3P binding protein DFCP1 consumes ATP to shape autophagosome formation, akin to how Dynamins perform membrane scission through GTP hydrolysis. DFCP1 is a highly-used imaging marker for following autophagosome formation, but very little is known about its function. Elucidation of how DFCP1 works at the molecular level will be invaluable to the autophagy field. The model put forward by the authors is intriguing, but the experiments need to be better designed to support the claims made in the manuscript.

Major points:

- 1) In Fig. 1, the authors measured DFCP1 ATP-hydrolysis without PI3P-rich membranes. This design begs the question- is the activity measured related to DFCP1's function on PI3P-rich subdomains of the ER? Based on the model presented in Extended Fig. 9, it appears that DFCP1 should only drive ATP hydrolysis during membrane constriction?

We thank the reviewer for raising this important point. First, we would like to clarify that these measurements were performed using the isolated N-terminus of DFCP1, which contains the NTPase domain but lacks membrane-binding domains.

To address the question if ATPase activity is increased in the presence of membranes, we purified full-length DFCP1 from insect cells and then tested its ATPase activity in the presence or absence of liposomes. Importantly, we found that the presence of liposomes could significantly increase DFCP1 ATPase activity (New Fig2c).

Interestingly, the ATPase activation by membranes seemed to be independent of the presence of PI3P. It is thus likely that PI3P serves as a recruiter of DFCP1 to membranes rather than stimulating its ATPase

activity. The early recruitment of DFCP1 to forming omegasomes suggests that it might have additional functions than the ATPase mediated constriction at a later stage.

- 2) In Fig. 2b, the authors imaged mNG-DFCP1 and mCh-p62 and suggested that DFCP1 couples ATP hydrolysis to membrane constriction. While the data can be used to phenomenologically show that there are three “phases” (initiation, maturation, and constriction) in omegasome (DFCP1-labeled) mediated autophagosome formation, it is difficult to see how one can clearly define the boundaries between the three phases in the images (e.g., why is the boundary between the initiation and maturation phases at $t=80s$ in the example shown?). Furthermore, it seems that the images were manually inspected and assigned. This method of quantification can make the results presented in Fig. 2c unreliable.

Our definition of the different phases of omegasome formation is based on manual inspection of a large number of movies by several highly trained researchers. However, we agree with the reviewer that the analysis would be strengthened by an automated analysis. In addition to our previous manual quantification of omegasome formation and constriction, we have therefore developed a script for automated quantification and used this to analyze an entire new data set (new Fig.3i). Importantly, the automated quantification confirms the data generated by the manual assessment: The persistence of DFCP1-labeled omegasomes is longer in the DFCP1 mutants, and the time from the largest ring-shape to the complete disappearance of the DFCP1 signal is delayed in the mutants, in line with the manual assessment of the delayed constriction phase.

- 3) Overexpression of DFCP1 is known to block autophagosome formation (PMID: 18725538). This observation means that alteration of DFCP1 levels will lead to changes in the dynamics of autophagosome formation. In this paper, the authors used stable cell lines expressing lower levels of mNG-DFCP1 (WT, K193A, and T189A) for imaging. The authors should carefully quantify and list the expression levels of DFCP1 in these lines (shown in Extended Fig. 3c), as they are used to report minute changes in omegasome formation time in Fig. 2c (from $t\sim 350s$ for WT to $t=400-500s$ for the mutants).

This is an absolutely valid point. Following transduction of cells with DFCP1 lentivirus, we have FACS-sorted all cell lines used to achieve the lowest DFCP1 expression level possible. With this, we have reached a DFCP1 level 3x over endogenous level. We have included the quantifications in New Extended Data Fig.3d, h. The ability of the stably expressed DFCP1 WT to rescue the loss of endogenous DFCP1 in the KO and KD systems indicates that the protein is functional. Importantly, the expression level is similar for DFCP1 WT and mutants, thus making them comparable in the various analyses.

- 4) From Figs. 1-3 the authors utilized cells under starvation (to induce autophagy) to assay for the functions of DFCP1. However, it has been shown that DFCP1 depletion does not affect the autophagy flux during starvation-induced autophagy (PMID: 18725538). The get out of this apparent paradox, the authors between LINE 186-201 appeared to invoke the argument that only a small fraction of autophagosomes formed during starvation come out of DFCP1-labeled omegasomes. The authors need to directly demonstrate this claim as it does not match the results from the existing literature.

In line with the existing literature (Axe et al., JCB 2008), we did not find a function for DFCP1 in starvation induced bulk autophagy as measured by LC3 recruitment to membranes (Fig. 4g) or by the cytosolic mKeima reporter (Fig.5a, Extended Data Fig.8a).

The reviewer questions our use of starvation as an inducer of selective autophagy. In addition to being a strong inducer of bulk autophagy, starvation has been shown to induce selective autophagy (Holdgaard et al., NatComm 2019) and EBSS-induced mitophagy (Allen GFG et al, EMBO Rep 2013), as has the inhibition of mTOR (An and Harper, 2008). In line with this, we observed that EBSS, Torin and DFP were equally strong inducers of mitophagy (Fig. 5f).

We agree with the reviewer that the argument that only a small fraction of autophagosomes come out of DFCP1 positive omegasomes does not match current literature, and by tracking movies with LC3B and DFCP1 we found that this argument indeed is not valid. How can we then explain the preferential requirement of DFCP1 for selective vs bulk autophagy?

We have performed automated analyses of maximum diameter vs duration of large numbers of omegasomes in cells expressing wild-type or ATP-binding/hydrolysis mutant DFCP1. Interestingly, these analyses showed that both small and large omegasomes form and close rapidly in cells expressing wild-type DFCP1, whereas large (but not small) omegasomes in cells expressing mutant DFCP1 show strongly delayed closure times (new Extended Figure 10e), This raises the possibility that DFCP1 ATPase only becomes essential when phagophores reach a certain size, as typical for selective autophagy, and that other ER remodelling factors are sufficient to sustain formation of smaller autophagosomes that mediate bulk autophagy. We thank the reviewer for raising this excellent point, which is now also discussed in the manuscript on page 12.

- 5) The authors suggested that DFCP1 helps the formation of autophagosomes around ubiquitinated substrates. Therefore, it is confusing why they looked at the role of DFCP1 in DFP-induced mitophagy. It is unclear whether ubiquitination is needed during DFP-induced mitophagy. It makes more sense for the authors to demonstrate how DFCP1 affects ubiquitin-dependent mitophagy.

Because we were unable to detect any direct interaction between DFCP1 and ubiquitin, we have chosen to remove the data and speculations on ubiquitin interactions. Instead, based on new tracking, we now think the preferential requirement of DFCP1 for selective autophagy may be related to omegasome size (see reply pt. 4).

- 6) The authors suggested that DFCP1 can directly bind to poly-ubiquitinated proteins, allowing it to mediate autophagosome formation around ubiquitinated substrates specifically. This claim is not sufficiently developed. One may also counter-argue that direct binding between DFCP1 and the substrates can impede autophagosome formation: LC3-labeled autophagosomes will need to compete with DFCP1 for substrate association.

As suggested by Reviewer #1, we have been investigating the binding of DFCP1 to ubiquitin in vitro. We have been unable to detect any direct interaction, so we have abandoned the idea that DFCP1 participates in recognition of ubiquitinated cargos. The reason why DFCP1 is required for selective but not for bulk autophagy thus remains unresolved, although we speculate that it might be related to differential size of the phagophore (see reply to pt. 4).

Reviewer #3:

In this manuscript, the authors demonstrate an important role of DFCP1 in autophagy as an ATPase, which relies on its ability to bind and hydrolyse ATP. Specifically, DFCP1 regulates the formation of autophagosomes. The DFCP1 mutants with reduced ATP binding or hydrolysis delay the release of nascent autophagosomes from omegasomes, resulting in p62 accumulation. Multiple advanced techniques such as super-resolution imaging have been employed to comprehensively study the new role of DFCP1. The paper is well written. Several major issues have to be addressed.

- 1) The authors define several stages of omegasome dynamics mainly based on the SIM images. The choice was made by the eyes, which would be very arbitrary. There should be a more quantitative or definitive way to segregate these stages. At least, the authors should show more representative SIM images with clear indicated features for staging.

We agree with the reviewer that the analysis would be strengthened by an automated analysis. In addition to our previous manual quantification of omegasome formation and constriction, we have therefore developed a script for automated quantification and used this to analyze an entire new data set (New Fig.3i). Importantly, the automated quantification confirms the data retrieved by the manual assessment: The persistence of DFCP1-labeled omegasomes is longer in the DFCP1 mutants, and the time from the largest ring-shape to the complete disappearance of the DFCP1 signal is delayed in the mutants, in line with the manual assessment of the delayed constriction phase.

- 2) For to the “initiation” stage, “characterized by a DFCP1 spot which is formed de novo, and to which p62 and LC3B are recruited a few seconds later”, the authors missed one critical parameter about the initiation, which is the successful rate of recruiting p62 and LC3B. Clearly, not every DFCP1 can lead to the formation of omegasome. Therefore, this successful rate would be very important to evaluate the function of DFCP1, especially, when the authors couldn't see a change on initiation and maturation for DFCP1 K193A or T189A.

To address this important question, we have tracked omegasome formation using mNG-DFCP1 and mCh-p62, followed by automated image analysis. The DFCP1 signal has been segmented, and p62 intensity within this segmentation has been recorded. Approximately 20% of newly forming omegasomes do not acquire p62, and there did not seem to be a difference if this was WT or mutant DFCP1 (New Fig.3e, f).

In addition, we have blindly tracked forming DFCP1 dots developing into omegasomes and determined the fluorescent intensity values for LC3 at these spots. If the intensity was 1.5x over background, we annotated it as a successful recruitment, all events under as a negative event (New Extended Data Fig.4d). In this analysis approximately 20 - 30% of newly forming omegasomes did not acquire LC3.

However, we cannot rule out that LC3 or p62 could have been under detection limit at the negative omegasomes.

- 3) The DFCP1 K193A and T189A mutants only delay omegasome formation. Basically, functional omegasomes can still form for autophagy. Therefore, it's not conclusive to say that delayed omegasomes without defects are directly related to p62 accumulation unless the number of omegasome formation for these two mutants is similar to what wild-type does.

We agree with the reviewer that the difference in the number of omegasomes between WT and the two mutants could in principle influence the interpretation of the p62 accumulation. We do, however, provide several lines of evidence that the increased number of omegasomes and p62 levels in the mutants are caused by their delayed constriction, and not an increased initiation of omegasomes:

First, we show that the early omegasome marker WIPI2, which is recruited to forming omegasomes and dissociates before they are fully constricted (New Extended Data Fig.5a), is not affected in the mutants (Fig.4c-f, Extended Data Fig.5b). This indicates that the onset of omegasome formation is not increased in the mutants compared to WT.

Second, omegasomes do form when ATP binding or hydrolysis is compromised, however, our manual analysis of omegasome dynamics show that the omegasome lifetime is increased, due to a delayed constriction. We now confirm this finding by automated analysis (New Fig.3i). Our automated analysis further showed that p62 closely follows DFCP1 (New Fig.3e).

Third, measuring autophagic flux by use of p62 and treatment or not with Bafilomycin A1 showed that the accumulation of p62 is not due to increased initiation of autophagy but rather a delayed degradation (Fig. 4i).

Thus, we believe that our conclusion that p62 accumulates in the mutants due to their delayed constriction is valid and supported by the presented data.

In addition, we have re-named the first phase of the omegasome sequence from 'initiation' to 'expansion'. 'Initiation' could be misleading, as it was not meant to imply the first appearance of the DFCP1 dot.

- 4) In order to conclude that the interaction between DFCP1 and ubiquitin would be influenced by the mutants K193A or T189A. More quantification should be conducted. For instance, a co-IP experiment similar to Fig. 4i should be performed.

As suggested by Reviewer #1, we have been investigating the binding of DFCP1 to ubiquitin in vitro. We have been unable to detect any direct interaction, so we have abandoned the idea that DFCP1 participates in recognition of ubiquitinated cargos. The reason why DFCP1 is required for selective but not for bulk autophagy thus remains unresolved, although we speculate that it might be related to differential size of the phagophore.

- 5) In the discussion section, the authors speculate that DFCP1 works on the membrane to regulate omegasome, similar what dynamin does. Dynamin is known for membrane tubulation [<https://www.nature.com/articles/nrm1313>]. Therefore, the authors should check the capability of DFCP1 and its mutants on membrane tubulation. This would significantly enhance the impact of this paper.

We have performed several attempts to measure membrane deformation by recombinant full-length DFCP1 on liposomes using negative staining EM. However, so far, we have not succeeded to show clear membrane deformation by DFCP1. This could be due to various reasons. First, our membrane composition might be suboptimal and not reflect the true omegasome membrane composition (although we observed a modest stimulation of ATPase activity by the same liposomes). Alternatively, DFCP1 could require a cellular co-factor (similar to amphiphysin in the case of Dynamin) to be fully active. Finally, the role of

DSCP1 could not be pure membrane deformation (although our phenotypes would hint at such a function), but rather have a different function, e.g. membrane tethering similar to Atlastin.

REVIEWER COMMENTS

Reviewer #1 (Remarks to the Author):

The authors have convincingly answered all my concerns and comments. I support publication and congratulate the authors on the nice work.

Reviewer #2 (Remarks to the Author):

In this revision, the authors have significantly revised their model on why DFCP1 is required for 'selective' autophagy. They have abandoned the thinking that DFCP1 acts by binding Ubiquitin. Instead, they propose that DFCP1's ATPase activity is required to constrict larger omegasomes (formed during the engulfment of large substrates during selective autophagy). The proposal is intriguing, but more data are needed to substantiate the model.

Major comments:

-In Fig. 3i, the authors show that omegasomes took the same amount of time to grow from 0.5 to ~1.2 um in cells expressing either WT DFCP1, K193A DFCP1, or K189A DFCP1. It also took the same time for omegasomes to shrink from ~1.2 um back to 0.5 um for the three types of cells. The main difference observed for the three types of cells is the time omegasomes stayed at ~0.5 um before their disappearance. How does this support the idea that DFCP1's ATPase activity is required to constrict larger omegasomes (Given that the activity apparently doesn't affect the omegasomes going from ~1.2 um back to 0.5 um)?

-In the Extended Fig. 10e, it appears that the DFCP1 mutants are significantly faster in constricting smaller omegasomes (for those <1.0 um), which contributes to the increased slope observed. On the other hand, if one were to extrapolate the straight lines drawn in the plots, it appears that the DFCP1 mutants may not be slower in constricting larger omegasomes (e.g., at 1.5 um) as compared to WT (the T189A mutant even seems to be faster). This differs from the authors' conclusions in LINES 323-326 in the main text. Also, this data should be presented in the main figures as opposed to the Supplementary info, as this is a major conclusion the authors draw in the paper.

-It is very nice that the authors have now developed an automated script to analyze omegasome expansion and constriction in the time-lapse images. The authors define that constriction starts when

omegasomes are at their maximal size and ends when omegasomes disappear. This assumption should be clearly stated in the text, as a change in definition could very well lead to changes in the conclusions drawn in the paper.

Reviewer #3 (Remarks to the Author):

All my concerns have been addressed.

RESPONSE TO REVIEWER COMMENTS

Reviewer #1 (Remarks to the Author):

The authors have convincingly answered all my concerns and comments. I support publication and congratulate the authors on the nice work.

Reviewer #2 (Remarks to the Author):

In this revision, the authors have significantly revised their model on why DFCP1 is required for 'selective' autophagy. They have abandoned the thinking that DFCP1 acts by binding Ubiquitin. Instead, they propose that DFCP1's ATPase activity is required to constrict larger omegasomes (formed during the engulfment of large substrates during selective autophagy). The proposal is intriguing, but more data are needed to substantiate the model.

Major comments:

-In Fig. 3i, the authors show that omegasomes took the same amount of time to grow from 0.5 to ~1.2 um in cells expressing either WT DFCP1, K193A DFCP1, or K189A DFCP1. It also took the same time for omegasomes to shrink from ~1.2 um back to 0.5 um for the three types of cells. The main difference observed for the three types of cells is the time omegasomes stayed at ~0.5 um before their disappearance. How does this support the idea that DFCP1's ATPase activity is required to constrict larger omegasomes (Given that the activity apparently doesn't affect the omegasomes going from ~1.2 um back to 0.5 um)?

We thank the reviewer for excellent comments! The reviewer is correct that during the formation of a new omegasome, the expansion and the *initial* constriction are only weakly affected by mutations in the ATPase domain. Indeed, the strongest effects occur during the final phase of constriction, and this is likely where the majority of phenotypes arise from. One potential explanation would be that both formation and the initial constriction are independent of DFCP1 (or at least its ATPase activity) and that DFCP1 is only involved in the final constriction phase.

How does this relate to the defects in selective autophagy that we observed? We agree with the reviewer that it is probably not a pure function of the size of the omegasome, as omegasome can still partially constrict. However, we observed a delay in the expansion of the omegasome from start to maximum diameter, especially in the ATP-binding defective K193A mutant - WT: 188 s +/- 9s (95% CI), T189A: 170 s +/- 12s (95% CI), K193A: 235s +/- 24s (95%CI). This suggests that nucleotide binding could also be required during the initial expansion of new omegasomes.

Moreover, a detailed analysis of the dataset presented in Fig. 3f (new Fig. 3g) shows that constriction – especially in cells expressing the ATP-binding defective mutant – was slower and stalled at a larger omegasome diameter. Moreover, these defects were more pronounced during the constriction of large omegasomes (Fig. 3g, Supplementary Fig. S4b), whereas constriction of both small and large omegasomes occurred rapidly in cells expressing WT DFCP1 (new Fig. 3g).

In the Extended Fig. 10e, it appears that the DFCP1 mutants are significantly faster in constricting smaller omegasomes (for those $<1.0 \mu\text{m}$), which contributes to the increased slope observed. On the other hand, if one were to extrapolate the straight lines drawn in the plots, it appears that the DFCP1 mutants may not be slower in constricting larger omegasomes (e.g., at $1.5 \mu\text{m}$) as compared to WT (the T189A mutant even seems to be faster). This differs from the authors' conclusions in LINES 323-326 in the main text. Also, this data should be presented in the main figures as opposed to the Supplementary info, as this is a major conclusion the authors draw in the paper.

Based on the concerns of this reviewer, we re-analyzed our datasets and provide now alternative plots which provide a clearer picture than the pure correlation plots. These plots are shown as new Fig. 3g and Supplementary Fig. S4a and b.

For this analysis, we classified individual omegasomes based on their maximum diameter or lifetime (new Fig. 3g, Supplementary Fig. S4b) and plotted their dynamics. Moreover, we analyzed the overall lifetime of each omegasome (new Supplementary Fig. S4a). Based on these analyses, we made the following observations:

Omeasomes in cells expressing WT DFCP1 showed little variation and a very tight distribution of the duration of the omegasome formation and constriction (new Supplementary Fig. S4a).

There was a clear correlation that smaller omegasomes were closed faster and large omegasomes took longer to constrict. However, all observed events fell within a small range (Fig. 3g, Supplementary Fig. S4a, 4b) and even the largest omegasomes closed within 450s of their formation.

In contrast, omegasomes in cells expressing ATPase mutants did not show this highly ordered behavior. Especially omegasomes in cells expressing the nucleotide-binding defective K193A mutant showed a strong increase in omegasome lifetimes. There were no short-lived omegasomes, suggesting that all omegasomes were delayed in their constriction (Supplementary Fig. S4a, b). Large omegasomes took significantly longer to constrict than the corresponding omegasomes in cells expressing WT DFCP1 (Fig. 3g, Supplementary Fig. S4b).

The ATPase hydrolysis defective mutant showed less severe phenotypes, but still showed an increase in omegasome lifetime and delays in closing large omegasomes. As this mutation retained ~20% residual ATPase activity (Fig. 1k), it might also partially retain its biological function.

Taken together, DFCP1 ATP binding and hydrolysis are critical for the complete closure of omegasomes and thus drive the effective formation of new autophagosomes. The delay in closure could result in an ineffective sequestration of large cargos.

Based on this new analysis, we have rewritten the corresponding sections of the Results and Discussion.

-It is very nice that the authors have now developed an automated script to analyze omegasome expansion and constriction in the time-lapse images. The authors define that constriction starts when omegasomes are at their maximal size and ends when omegasomes disappear. This assumption should be clearly stated in the text, as a change in definition could very well lead to changes in the conclusions drawn in the paper.

As suggested, we have added a section describing our image analysis and definition to the methods part of the manuscript.

Reviewer #3 (Remarks to the Author):

All my concerns have been addressed.

REVIEWERS' COMMENTS

Reviewer #2 (Remarks to the Author):

The revised version has adequately tackled the raised points, making it fit for publication.